# Longitudinal map of transcriptome changes in the Lyme pathogen *Borrelia burgdorferi* during tick-borne transmission

**Anne L Sapiro[1]\*, Beth M Hayes[1], Regan F Volk[2], Jenny Y Zhang[1], Diane M Brooks[3], Calla Martyn[1], Atanas Radkov[1], Ziyi Zhao[1], Margie Kinnersley[3], Patrick R Secor[3], Balyn W Zaro[2], Seemay Chou[1]\***

[1]Department of Biochemistry & Biophysics, University of California, San Francisco, San Francisco, United States; [2]Department of Pharmaceutical Chemistry and Cardiovascular Research Institute, University of California, San Francisco, San Francisco, United States; [3]Division of Biological Sciences, University of Montana, Missoula, United States

**\*For correspondence:**
annesapiro@gmail.com (ALS);
seemaychou@gmail.com (SC)

**Abstract** *Borrelia burgdorferi* (*Bb*), the causative agent of Lyme disease, adapts to vastly different environments as it cycles between tick vector and vertebrate host. During a tick bloodmeal, *Bb* alters its gene expression to prepare for vertebrate infection; however, the full range of transcriptional changes that occur over several days inside of the tick are technically challenging to capture. We developed an experimental approach to enrich *Bb* cells to longitudinally define their global transcriptomic landscape inside nymphal *Ixodes scapularis* ticks during a transmitting bloodmeal. We identified 192 *Bb* genes that substantially change expression over the course of the bloodmeal from 1 to 4 days after host attachment. The majority of upregulated genes encode proteins found at the cell envelope or proteins of unknown function, including 45 outer surface lipoproteins embedded in the unusual protein-rich coat of *Bb*. As these proteins may facilitate *Bb* interactions with the host, we utilized mass spectrometry to identify candidate tick proteins that physically associate with *Bb*. The *Bb* enrichment methodology along with the ex vivo *Bb* transcriptomes and candidate tick interacting proteins presented here provide a resource to facilitate investigations into key determinants of *Bb* priming and transmission during the tick stage of its unique transmission cycle.

## eLife assessment

In this Tools and Resources article, the authors overcome the challenge of low *Borrelia burgdorferi* numbers during infection for analyses such as RNA-sequencing or mass spectrometry. They do so by physically enriching for spirochetes, which is **important**, as it provides technical advances for the study of global transcriptomic changes of *B. burgdorferi* during tick feeding, helping to build on the knowledge already collected by the field. The evidence presented is **compelling**, and the strategy described here could benefit researchers in the field and possibly also support broader applications.

## Introduction

Vector-borne microbial pathogens are transmitted by the bite of arthropods and have evolved sophisticated ways to adapt to vastly different environments as they move between vector and host. Uncovering their adaptive mechanisms can not only open avenues for disrupting pathogen transmission but

also provide fundamental insights into vector physiology and microbial symbioses (*Shaw and Catteruccia, 2019*). Lyme disease, the most reported vector-borne disease in North America, is caused by the bacterial pathogen *Borrelia burgdorferi* (*Bb*) (*Rosenberg et al., 2018*; *Steere et al., 2016*). Its primary vector, the blacklegged tick *Ixodes scapularis,* acquires and transmits *Bb* through two separate multi-day bloodmeals, one in which *Bb* is acquired from an infected vertebrate host and a second, during the subsequent life stage, in which *Bb* is transmitted to a new host (*Tilly et al., 2008*). During the transmission bloodmeal, *Bb* proliferates in the tick midgut before a subset of these cells disseminate to the salivary glands (*Dunham-Ems et al., 2009*). *Bb* is deposited into the new host via the tick saliva extruded into the bite site (*Ribeiro et al., 1987*). Given the prolonged nature of *I. scapularis* feeding, the high specificity of vector-pathogen relationships, and the complicated array of events needed for successful *Bb* transmission, the tick bloodmeal provides an opportune intervention point for preventing pathogen spread. However, we do not currently have a clear understanding of the molecular mechanisms involved in this process.

*Bb* must adapt to dramatically different environments as it cycles from tick to vertebrate host, and understanding the genes involved in this process will enable the identification of key interactions to target to prevent transmission. When an infected tick feeds, *Bb* responds to bloodmeal-induced environmental changes and undergoes cellular modifications driven by key transcriptional circuits, including the RpoN/RpoS sigma factor cascade and the Hk1/Rrp1 two-component system (*Radolf et al., 2012*). In vitro analyses of *Bb* cells cultured in tick- or mammal-like growth conditions have pointed to additional genetic determinants of tick-borne transmission by revealing widespread transcriptome remodeling during host switching (reviewed in *Samuels et al., 2021*). However, it is not fully clear how these in vitro expression changes correspond to complex in vivo changes over the course of a transmission bloodmeal. Capturing comprehensive, longitudinal data on *Bb* gene expression from inside its vector has been hampered by technical challenges due to the dynamic nature of the bloodmeal and the general low abundance of bacterial cells relative to the tick (*Samuels et al., 2021*). Some progress has been made in making transcriptome-wide measurements from the tick. *Bb* sequence enrichment coupled with microarrays identified large scale changes in *Bb* gene expression between the first and second tick bloodmeals (*Iyer et al., 2015*). More recently, enriching *Bb* sequences from infected tick RNA-seq libraries through TBDCapSeq has provided clearer resolution into differences in *Bb* gene expression as it cycles between the fed tick and mammalian host (*Grassmann et al., 2023*). Still, we have limited temporal resolution into the molecular transitions that happen across key steps of tick feeding. This problem necessitates novel approaches to capture the transcriptomic changes of *Bb* within the natural tick environment.

While the full landscape of *Bb* transmission determinants is not yet known, we do have a growing knowledge of functional processes that are critical during the tick stage, such as motility, metabolism, and immune evasion (*Kurokawa et al., 2020*; *Phelan et al., 2019*). These functions often rely on the unique protein-rich *Bb* outer surface. Notably, several specific tick–*Bb* protein–protein interactions are important for *Bb* survival, migration, or transmission to the next host. *Bb* encodes an extensive outer surface protein (Osp) family with members that are differentially expressed during host switching. One of these proteins, OspA, binds a tick cell surface protein, tick receptor for OspA (TROSPA), which is required for successful *Bb* colonization of the tick midgut during the first acquisition bloodmeal (*Pal et al., 2004*). Several other proteins have also been linked to *Bb* migration within the tick (*Pal et al., 2021*). For example, BBE31 binds a tick protein TRE31, and disruption of this interaction decreases the number of *Bb* cells that successfully migrate from the tick gut to salivary glands (*Zhang et al., 2011*). However, these interactions alone are not sufficient to block *Bb* growth or migration in ticks, suggesting there are likely additional molecular factors from *Bb* and ticks at play during tick-borne transmission.

To provide a more comprehensive set of *Bb* determinants driving tick-borne transmission of this important human pathogen, we developed a novel sequencing-based strategy for ex vivo transcriptomic profiling of *Bb* populations within infected nymphal *I. scapularis* ticks as they transmit *Bb* to a mouse host. We used this method to longitudinally map genome-wide *Bb* expression changes for bacterial cells isolated from ticks during the transmission bloodmeal from 1 to 4 days after attachment. We identified 192 highly differentially expressed genes, including genes previously implicated in *Bb* transmission as well as many others. Genes upregulated during tick transmission included many outer surface lipoproteins, suggesting *Bb* dramatically remodels its cell envelope as it migrates through the

tick. Mass spectrometry analyses revealed dramatic changes in the tick environment over feeding, identifying new potential determinants of a more extensive and diverse set of tick–microbe molecular interactions than previously appreciated. The *Bb* enrichment method and resulting datasets serve as a community resource to facilitate further investigations into the key determinants of *Bb* transmission.

## Results
### A two-step enrichment process facilitates robust transcriptional profiling of *Bb* during the tick bloodmeal

To gain a more comprehensive understanding of *Bb* gene expression throughout the tick phase of the transmission cycle, we developed an experimental approach to characterize the *Bb* transcriptome of spirochetes isolated from nymphal *I. scapularis* ticks during a days-long bloodmeal in which *Bb* is transmitted to a vertebrate host. We aimed to establish a longitudinal transcriptional profile encompassing key pathogen transmission events each day of feeding after ticks attached to their mouse bloodmeal hosts (*Figure 1A*). We fed *Bb*-infected nymphal ticks on naive mice and collected the feeding ticks at daily intervals after the start of feeding until 4 days after attachment, at which time the ticks had fully engorged and detached from the mice. The major bottleneck for such an RNA sequencing (RNA-seq) approach is capturing sufficient quantities of *Bb* transcripts from complex multi-organism samples in which pathogen transcripts represent a very small minority. Our initial attempts to uncover *Bb* mRNA by simply removing tick mRNA with polyA-depletion and removing tick rRNA sequences using Depletion of Abundant Sequences by Hybridization (DASH; *Dynerman et al., 2020*; *Gu et al., 2016*) were unsuccessful. This approach resulted in an average of only 0.09% of RNA-seq reads mapping to *Bb* mRNA – approximately 10-fold less than we estimated would be needed to feasibly obtain robust transcriptome-wide differential gene expression analysis (*Haas et al., 2012*).

To dramatically increase *Bb* transcript representation in our libraries, we physically enriched *Bb* cells from tick lysates prior to library preparation by adding an initial step of immunomagnetic separation (*Figure 1B*). We took advantage of a commercial antibody previously generated against whole *Bb* cells (α*Bb*, RRID: AB_1016668). By western blot analysis, we confirmed that α*Bb* specifically recognized several *Bb* proteins, including surface protein OspA (*Figure 1—figure supplement 1A*), which is highly prevalent on the *Bb* surface in the tick (*Ohnishi et al., 2001*). In addition, immunofluorescence microscopy with α*Bb* showed clear recognition of *Bb* cells from within the tick at each day of feeding (*Figure 1—figure supplement 1B*). After collecting infected nymphal ticks from mice one, two, three, and four days post-attachment, we used α*Bb* and magnetic beads to enrich *Bb* cells from the tick material in the lysates. We tracked relative *Bb* enrichment through RT-qPCR of *Bb flaB* RNA and tick *gapdh* RNA in the separated samples. Measuring *Bb flaB* RNA from both *Bb*-enriched samples and their matched *Bb*-depleted fractions, we found over 95% of total *Bb flaB* RNA was present in enriched fractions (*Figure 1C*), suggesting our approach captured the vast majority of *Bb* transcripts from the tick.

Total RNA recovered from the enrichment process was used to create RNA-seq libraries that subsequently underwent depletion of highly expressed rRNA sequences from tick, mouse, and *Bb* using DASH, which targets unwanted sequences for degradation by Cas9 (*Gu et al., 2016*; *Ring et al., 2022*). DASH reduced unwanted sequences from 94% to 9% of our total libraries, greatly increasing the relative abundance of *Bb* transcripts (*Figure 1D*). For resulting libraries generated across feeding, between 0.6% and 3.4% of reads mapped to annotated *Bb* genic regions (*Figure 1E*), which translated to an average of 4.3 million *Bb* genic reads per sample (*Figure 1F* and *Figure 1—source data 1*).

As expected, for samples pulled from the mice one, two, and three days after attachment, the majority of the remaining sequencing reads mapped to the *I. scapularis* genome (72–83%; *Figure 1—source data 1*). On day 4, when ticks were fully engorged and were recovered from mouse cages rather than pulled from the mice, we found that fewer reads mapped to the *I. scapularis* genome (24–44%). Only a small percentage of reads from all samples mapped to the host *Mus musculus* genome (1–3%). To identify the source of the remaining reads in the day 4 samples, we ran the data through a publicly available computational pipeline that identifies microbes in sequencing datasets, CZ ID (*Kalantar et al., 2020*). This analysis led us to discover that a large percentage of day 4 reads mapped to bacterial species *Pseudomonas fulva* (41–64%) (*Figure 1—source data 1*), which may have

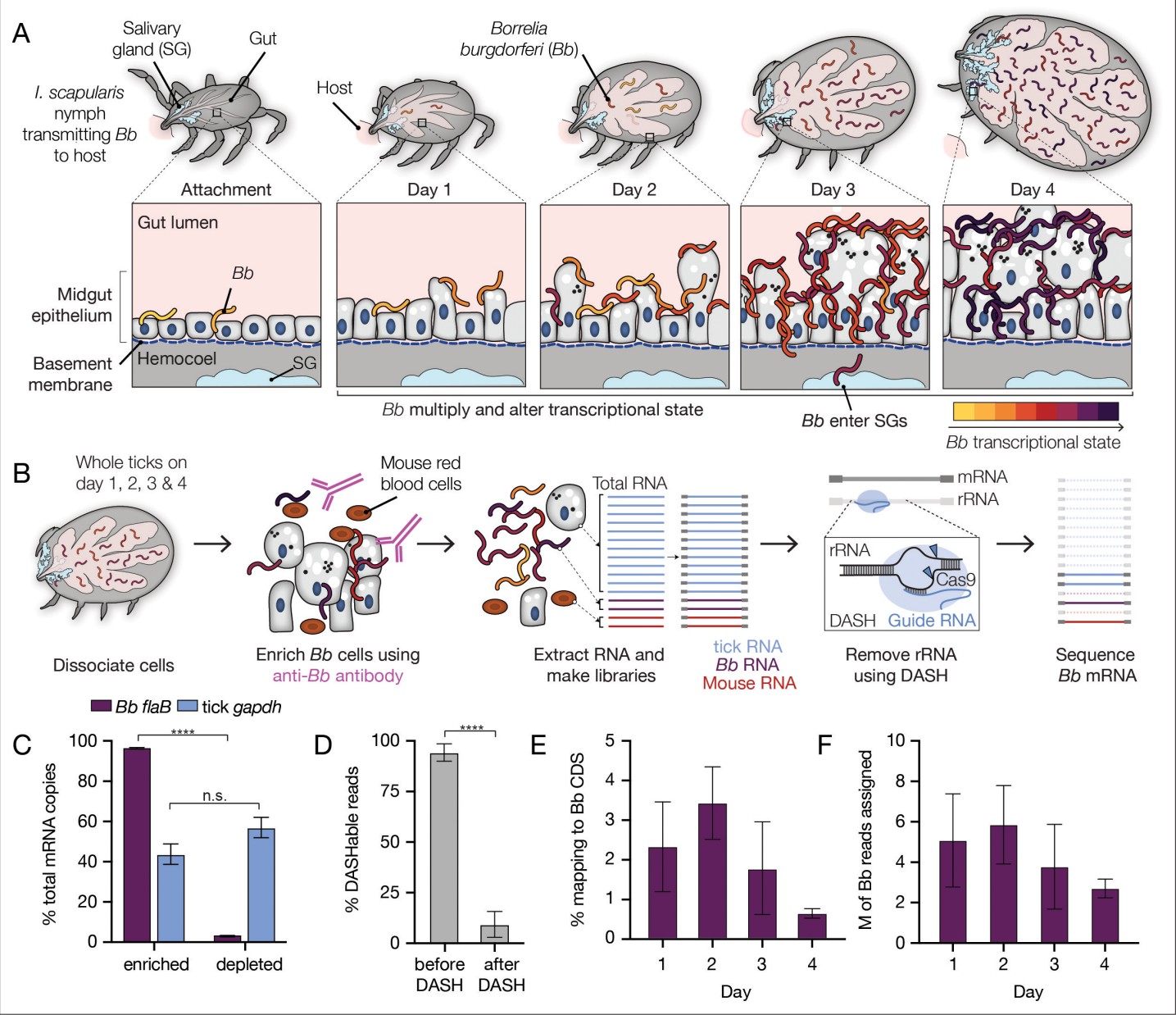

**Figure 1.** A two-step enrichment process facilitates robust transcriptional profiling of *Bb* during the tick bloodmeal. (**A**) Schematic of *Bb* during nymphal *I. scapularis* feeding. *Bb* in the nymphal tick midgut respond to the nutrient-rich bloodmeal by multiplying and changing their transcriptional state (**Ouyang et al., 2012**; **de Silva and Fikrig, 1995**). At the same time, the tick gut undergoes numerous changes to digest the bloodmeal (**Caimano et al., 2015**; **Sonenshine and Anderson, 2014**). After two to three days of feeding, a small number of *Bb* leave the midgut and enter the salivary glands (blue), while the majority are left behind in the gut after engorgement (**Dunham-Ems et al., 2009**). (**B**) Schematic of *Bb* enrichment process from feeding ticks. Whole ticks are dissociated, α*Bb* antibodies are added to lysates, and antibodies and *Bb* are captured magnetically. RNA is extracted and RNA-seq libraries are prepared. DASH is then used to remove rRNA before sequencing. This process increases *Bb* reads in the resulting sequencing data. (**C**) RT-qPCR results showing the percentage of *Bb flaB* and *I. scapularis gapdh* RNA in the enriched versus depleted fractions after the enrichment process. Data come from 4 replicates each from day 2, day 3, and day 4, mean +/-SE. ****p-value <0.0001, paired t test. Nearly all *Bb flaB* RNA was found in the enriched fraction. (**D**) The percentage of reads mapping to rRNA before and after DASH. n=4. Data are shown as mean +/-SD. ****p-value <0.0001, paired t test. rRNA reads are drastically reduced after DASH. (**E**) The percentage of reads in RNA-seq libraries mapping to *Bb*. *Bb* mRNA reads make up a larger proportion of libraries than without enrichment. n=4. Data are shown as mean +/-SD, see *Figure 1—source data 1*. (**F**) The number of reads in millions (M) mapped to *Bb* for each day. n=4. Data are shown as mean +/-SD. An average of 4.3 million reads per sample mapped to *Bb* genes, covering 92% of annotated genes with at least 10 reads.

The online version of this article includes the following source data and figure supplement(s) for figure 1:

**Source data 1.** Overview of mapping statistics from 16 *Bb* sequencing samples.

*Figure 1 continued on next page*

*Figure 1 continued*

**Figure supplement 1.** α*Bb* antibody recognizes OspA and binds *Bb* in the tick throughout the bloodmeal.

**Figure supplement 1—source data 1.** Full unedited original western blot of α*Bb* on lysate from cultured *Bb*.

**Figure supplement 1—source data 2.** Labeled uncropped western blot of α*Bb* on lysate from cultured *Bb*: wildtype (A3), a mutant lacking *ospA* (ospA1), and the mutant with *ospA* restored, *ospB* restored, and *ospC* restored. Molecular weight markers are shown on left (kD), and OspA size is noted on right.

**Figure supplement 2.** Enrichment process does not induce large scale gene expression changes in in vitro cultured *Bb*.

**Figure supplement 2—source data 1.** Differential expression analysis results between in vitro cultured *Bb* before and after *Bb* enrichment.

been present in our mouse cages. While these samples had a lower percentage of reads that mapped to *Bb*, broad transcriptome coverage was still obtained by increasing total sequencing depth. Across all samples from the 4 days, at least 10 reads mapped to 92% of annotated *Bb* protein coding genes and pseudogenes. The median number of reads per gene in each sample varied from 338 to 1167 reads (*Figure 1—source data 1*). This coverage was sufficient for statistically significant downstream differential expression analyses for the vast majority of *Bb* genes.

To evaluate whether our approach introduced any major artifacts in *Bb* expression, we sequenced and compared RNA-seq libraries from in vitro cultured *Bb* cells before and after immunomagnetic enrichment. We found minimal expression differences (29 genes with p<0.05, fold changes between 0.83 and 1.12; *Figure 1—figure supplement 2* and *Figure 1—figure supplement 2—source data 1*), suggesting experimental enrichment did not significantly alter global transcriptome profiles for *Bb*. Thus, our enrichment approach enabled genome-wide analysis of *Bb* population-level expression changes that occur within the feeding nymph as *Bb* is transmitted to the host.

## Global ex vivo profiling of *Bb* reveals extent and kinetics of transcriptional changes

To provide a broad overview of *Bb* expression changes in the tick during the nymphal *I. scapularis* transmission bloodmeal, we performed principal component analysis (PCA) on the *Bb* transcriptome data from one, two, three, and four days after attachment (n=4). We reasoned that if many longitudinal expression changes were occurring across *Bb* populations, we would observe greater data variability between time points than between biological replicates. Indeed, we found replicates from each day grouped together, whereas distinct time points were largely non-overlapping. The first principal component, which explained 64% of the variance in our data, correlated well with day of feeding (*Figure 2A*). The global pattern suggested that *Bb* gene expression changes generally trended in the same direction over the course of feeding with the most dramatic differences between flanking timepoints on day 1 and day 4.

Using day 1 (early attachment) as a baseline, we performed differential expression analysis for all *Bb* genes at subsequent time points (day 2, day 3, day 4; *Figure 2B* and *Figure 2—source data 2*). We examined changes with p-values <0.05 when adjusted for multiple hypothesis testing and fold changes above a twofold threshold (listed in *Figure 3—source data 1*). These analyses mirrored the global longitudinal expression pattern predicted by the PCA. The total number of differentially expressed (DE) genes when compared to day 1 increased with each subsequent timepoint to day 4. By day 4, there were 186 DE genes, including 153 upregulated and 33 downregulated (*Figure 2C–E*). Across all later time point comparisons to day 1, DE genes were highly overlapping and largely changed in the same directions. For example, of the DE genes that increased on day 2, 29 of 30 were still increased on day 3, and 29 of 30 were still increased on day 4. In the day 2, day 3, and day 4 comparisons to the day 1 baseline, we found 192 DE genes in total (*Figure 3—source data 1*). We observed some differences between gene expression patterns, such as the overall timing and kinetics of expression changes. Transcript levels for some DE genes changed suddenly over the course of feeding, while others were more gradual. To our knowledge, this is the first comprehensive report of global *Bb* expression changes over multiple stages of a tick feeding.

To assess the integrity of our dataset, we first examined expression profiles of previously characterized targets of major transcriptional programs activated at the onset of the bloodmeal: the RpoN/RpoS sigma factor cascade and the Hk1/Rrp1 two-component system (*Caimano et al., 2015*; *Grassmann et al., 2023*). Between day 1 and day 4, the expression of *rpoS* increased (*Figure 2—figure*

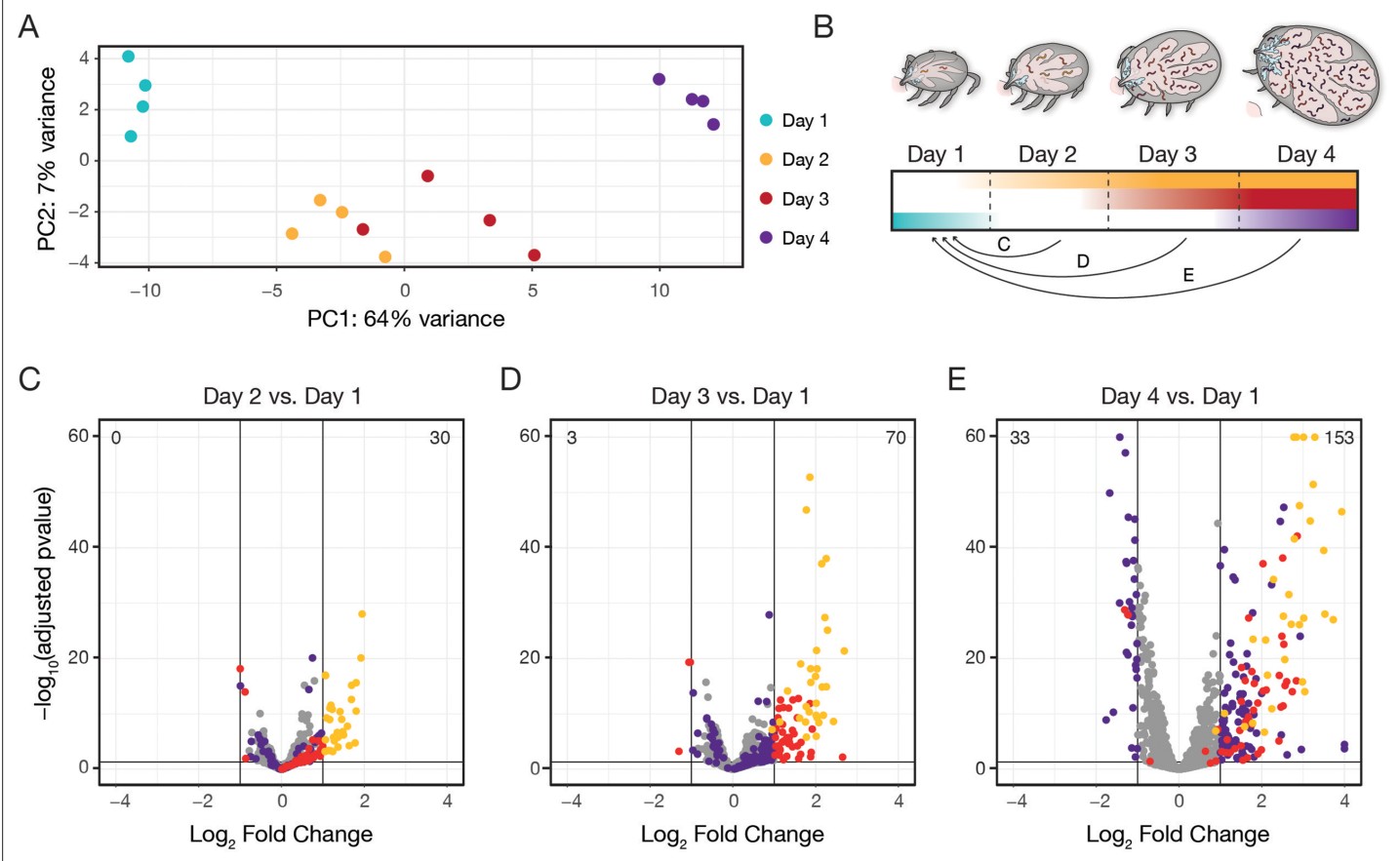

**Figure 2.** Global ex vivo profiling of *Bb* reveals extent and kinetics of transcriptional changes. (**A**) Principal component analysis of normalized read counts from samples from across feeding, see *Figure 2—source data 1*. PC1 correlates strongly with day of feeding. (**B**) Schematic depicting how data was analyzed, as pairwise comparisons between the first day after attachment and all other days. (**C–E**) Volcano plots of differentially expressed genes comparing day 2 versus day 1 (**C**), day 3 versus day 1 (**D**), and day 4 versus day 1 (**E**). The total number of upregulated genes is shown in the top right and the number of downregulated genes is shown in the top left. Yellow dots are genes that first change expression between day 1 and day 2, red dots are genes that first change expression between day 1 and day 3, and purple dots are genes that first change expression between day 1 and day 4. Two genes with log$_2$ fold changes >4 are shown at x=4, and five genes with -log$_{10}$(padj)>60 are shown at y=60. Only genes with p-value <0.05 from Wald tests and at least a twofold change are highlighted, see *Figure 2—source data 2*. n=4. By day 4 of feeding, 153 genes are upregulated and 33 genes are downregulated from day 1 baseline levels.

The online version of this article includes the following source data and figure supplement(s) for figure 2:

**Source data 1.** DESeq2 normalized counts for all genes across all samples.

**Source data 2.** Transcriptome-wide differential expression analysis results from *Bb* across tick feeding timepoints.

**Figure supplement 1.** Ex vivo RNA-seq corroborates transcriptional programs in the tick.

**Figure supplement 2.** Genes changing over tick feeding overlap with genes that change expression in previously probed tick feeding contexts.

supplement 1A), as expected (*Hübner et al., 2001*), and the majority of genes activated by RpoS in the feeding tick (*Grassmann et al., 2023*), including canonical targets *ospC* and *dbpA*, were also significantly upregulated (79/89, p<0.05, Wald tests; *Figure 2—figure supplement 1B*). The majority of genes activated or repressed by Rrp1 in vitro (*Caimano et al., 2015*) also trended in the expected direction ex vivo between day 1 and day 4 (111/148 upregulated, 37/57 downregulated, p<0.05, Wald tests, *Figure 2—figure supplement 1C*). We also examined the expression trends of genes regulated by Rel$_{Bbu}$ as part of the stringent response, another major transcriptional program active in *Bb* in the tick during nutrient starvation (*Drecktrah et al., 2015*). About half of Rel$_{Bbu}$-regulated genes changed in the direction expected if the stringent response was active during this time (129/251 upregulated, 111/226 downregulated, p<0.05, Wald tests) (*Figure 2—figure supplement 1D*); however, more of

the genes that were twofold downregulated over feeding are regulated by Rel$_{Bbu}$ than either RpoS or Rrp1, suggesting it may play a role during this time frame.

We then compared the 192 twofold DE genes in our longitudinal time course to those identified in two previous studies that measured *Bb* gene expression changes from culture conditions approximating the unfed tick and fed tick through modulation of temperature and/or pH (*Ojaimi et al., 2003*; *Revel et al., 2002*). 31% of the DE genes upregulated from day 1 (49/158) were more highly expressed in 'fed tick' conditions compared to 'unfed tick' conditions in one or both studies, while 24% of the DE genes downregulated from day 1 (8/24) were more highly expressed in 'unfed tick' conditions in one or both studies (*Figure 2—figure supplement 2A* and *Figure 3—source data 1*). The studies become more concordant when focusing on the DE genes that were upregulated on day 2, which were generally the genes that changed the most dramatically in the time course. 70% of DE genes upregulated on day 2 (21/30) were more highly expressed in 'fed tick' conditions in these previous studies, suggesting that the majority of the most dramatic gene expression changes we saw across feeding agree with what was observed in these previous studies.

We also compared the DE genes to two studies that assessed *Bb* gene expression differences in fed nymphs versus dialysis membrane chambers (DMCs), which mimic *Bb* conditions in the mammal (*Grassmann et al., 2023*; *Iyer et al., 2015*). 63% of all upregulated DE genes (100/158) were differentially expressed between fed nymphs and DMCs in one or both studies (*Figure 2—figure supplement 2B* and *Figure 3—source data 1*). The genes that were more highly expressed in nymphs were most concentrated amongst the day 2 DE genes (17/30, 57%), while the genes more highly expressed in DMCs were concentrated amongst the day 3 and day 4 DE genes (55/128, 43%). These comparisons suggested that the timing and magnitude of gene expression changes during feeding may indicate whether gene expression will peak in the tick or continue rising once *Bb* is transmitted to the host.

Through comparisons to these previous studies, we were able to verify that our data captured many expected transcriptional trends occurring during tick feeding. Nevertheless, 14% of the twofold

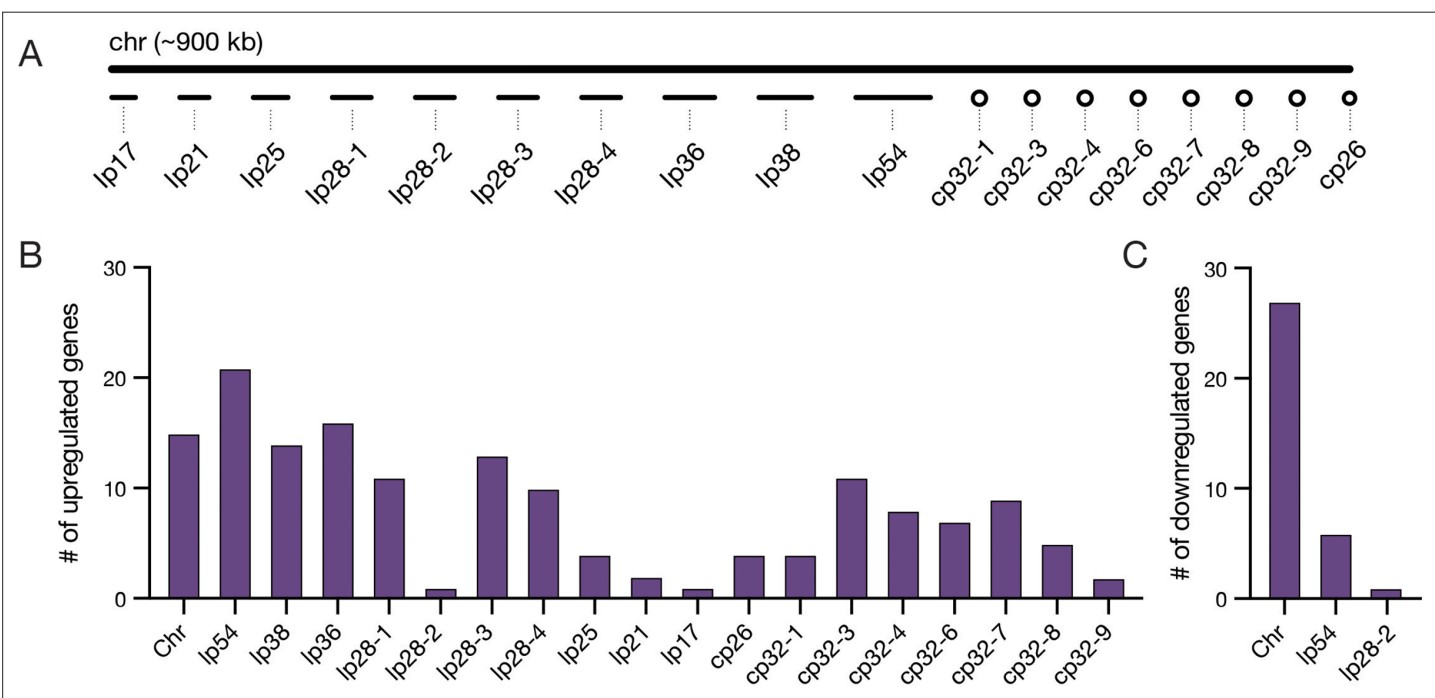

**Figure 3.** *Bb* genes upregulated during feeding are found predominantly on plasmids. (**A**) Schematic of the chromosome and plasmids in the *Bb* B31-S9 genome. Plasmid names denote whether the plasmid is linear (lp) or circular (cp) and the length of plasmids in kilobases (kb). For example, lp17 is a 17 kb linear plasmid. Genome is shown approximately to scale. (**B–C**) The number of genes from each chromosome or plasmid that increased (**B**) or decreased (**C**) expression twofold during feeding, see *Figure 3—source data 1* for gene information. Upregulated genes are distributed across plasmids, while most downregulated genes are found on the chromosome and lp54.

The online version of this article includes the following source data for figure 3:

**Source data 1.** Twofold differentially expressed *Bb* genes from across tick feeding timepoints.

DE genes were not previously found to change expression in these different tick-feeding contexts (*Grassmann et al., 2023*; *Iyer et al., 2015*; *Ojaimi et al., 2003*; *Revel et al., 2002*) or identified in these RNA-seq studies as dependent upon RpoS, Rrp1, or Rel$_{Bbu}$ (*Caimano et al., 2015*; *Drecktrah et al., 2015*; *Grassmann et al., 2023*), which are three known *Bb* regulatory programs active in the tick (*Samuels et al., 2021*). These additional genes highlight the necessity of measuring transcription in the tick environment and suggest we uncovered gene expression changes specific to the tick stage of the *Bb* enzootic cycle. The nature and dynamics of these changes provide insights into potential genetic determinants of *Bb* survival, proliferation, and dissemination in the tick during transmission.

## *Bb* genes upregulated during feeding are found predominantly on plasmids

*Bb* has a complex, highly fragmented genome (*Barbour, 1988*; *Figure 3A*), including numerous plasmids that are necessary during specific stages of the enzootic cycle (*Schwartz et al., 2021*) suggesting they contain genes that are crucial for pathogen transmission and survival. In fact, many genes found on the plasmids have been previously shown to alter expression upon environmental changes or in different host environments (*Iyer et al., 2015*; *Ojaimi et al., 2005*; *Revel et al., 2002*; *Tokarz et al., 2004*). Thus, we reasoned that many of the 192 DE *Bb* genes that change expression from day 1 to any later feeding time point (*Figure 3—source data 1*) would reside on the plasmids, and we examined their distribution throughout the genome. Consistent with these previous reports, we found that most of the upregulated genes were located on the plasmids (143/158; 90%), while fewer were found on the chromosome (15/158; 10%; *Figure 3B*), which is home to the majority of metabolic and other housekeeping genes. In contrast, the majority of the downregulated genes were found on the chromosome (27/34, 79%; *Figure 3C*).

Several plasmid-encoded genes that were longitudinally upregulated in our dataset have known roles during the tick bloodmeal or in mammalian infection. Linear plasmid 54 (lp54), which is an essential plasmid present in all *Bb* isolates (*Casjens et al., 2012*), contained the largest number of upregulated genes. Many of the genes on lp54 are regulated by RpoS during feeding, including those encoding adhesins DbpA and DbpB, which are important for infectivity in the host (*Blevins et al., 2008*). This set also included five members of a paralogous family of outer surface lipoproteins BBA64, BBA65, BBA66, BBA71, and BBA73. BBA64 and BBA66 are necessary for optimal transmission via the tick bite (*Gilmore et al., 2010*; *Patton et al., 2013*). These findings indicate our dataset captures key *Bb* transcriptional responses known to be important for survival inside the tick during a bloodmeal.

Many upregulated genes were also encoded by cp32 plasmid prophages. *Bb* strain B31-S9 harbors seven cp32 isoforms that are highly similar to each other (*Casjens et al., 2012*). When cp32 prophages are induced, phage virions called φBB1 are produced (*Eggers and Samuels, 1999*). In addition to phage structural genes, cp32 contain loci that encode various families of paralogous outer surface proteins (*Stevenson et al., 2000*). Amongst the cp32 genes that increased over feeding were members of the RevA, Erp, and Mlp families, which are known to increase expression during the bloodmeal (*Gilmore et al., 2001*). We also found several phage genes that were upregulated, including those encoding proteins annotated as phage terminases on cp32-3, cp32-4, and cp32-7 (BBS45, BBR45, and BBO44). Some cp32 genes have been shown to change expression in response to the presence of blood (*Tokarz et al., 2004*) and as a part of the stringent response regulated by Rel$_{Bbu}$ (*Drecktrah et al., 2015*), while BBD18 and RpoS regulate prophage production in the tick midgut after feeding (*Wachter et al., 2023*). Our data suggest that some prophage genes are upregulated over the course of tick feeding, raising the possibility that cp32 prophage are induced towards the end of feeding. Overall, our data support the long-held idea that the *Bb* plasmids, which house many genes encoding cell envelope proteins, proteins of unknown function, and prophage genes, play a critical role in the enzootic cycle during the key transition period of tick feeding.

## *Bb* genes encoding outer surface proteins are highly prevalent among upregulated genes

To gain a better overall sense of the types of genes that changed over feeding and the timing of those changes, we grouped DE genes into functional categories. Since a high proportion of plasmid genes encode lipoproteins within the unique protein-rich outer surface of *Bb*, genes of unknown function, and predicted prophage genes (*Casjens et al., 2000*; *Fraser et al., 1997*), we expected that many

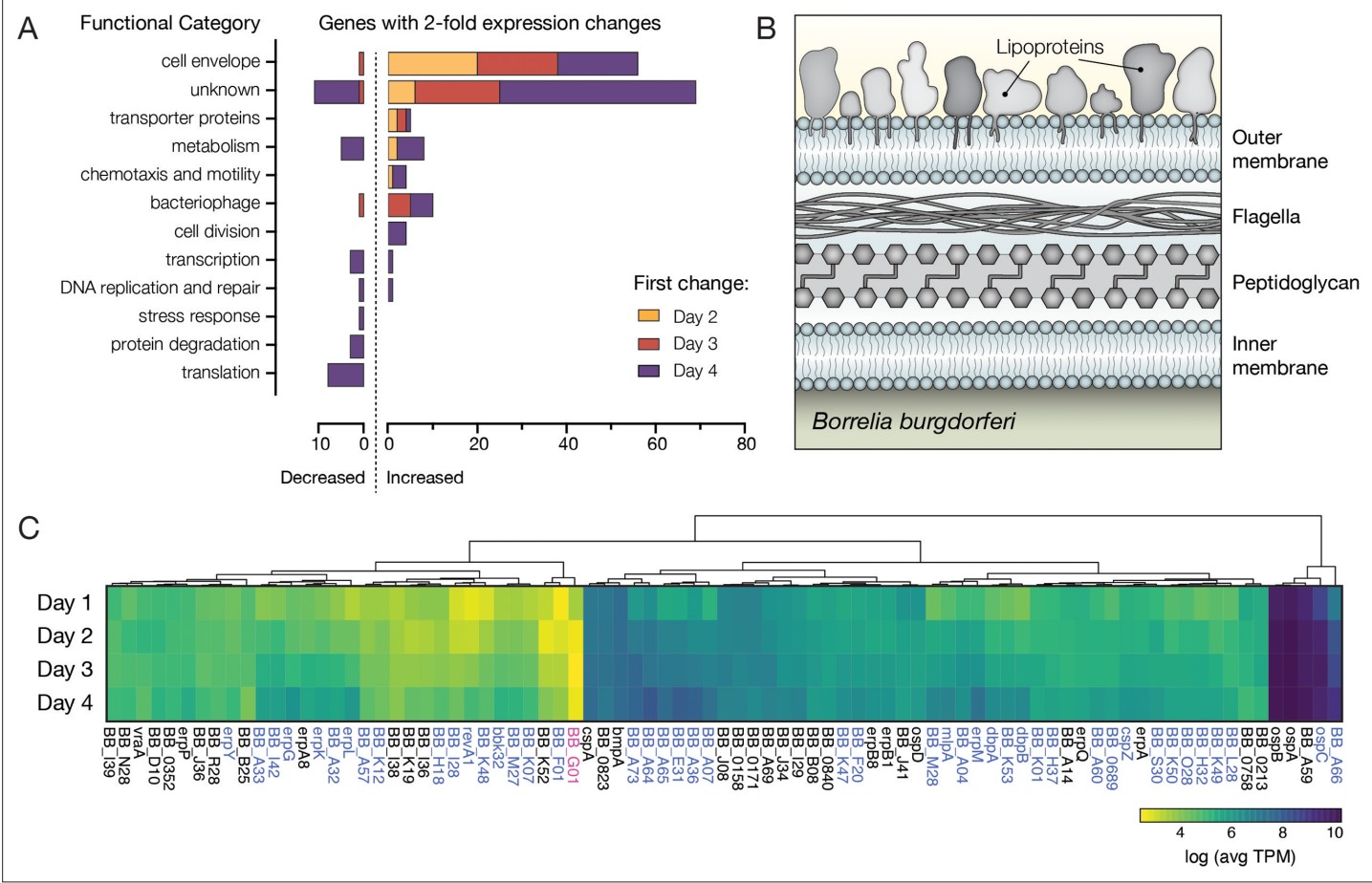

**Figure 4.** *Bb* genes encoding outer surface proteins are highly prevalent among upregulated genes. (**A**) The number of *Bb* genes that change over the course of tick feeding sorted into functional categories. Genes that first change 2 days after attachment are shown in yellow, 3 days after attachment in red, and 4 days after attachment in purple. A majority of upregulated genes fall into cell envelope and unknown categories. (**B**) Schematic of the outer membrane of *Bb* showing outer surface lipoproteins. Lipoproteins can also reside in the periplasmic space. (**C**) Heat map of expression levels of all genes encoding outer surface lipoproteins as average Transcripts Per Million (TPM) across the 4 days of tick feeding, see *Figure 4—source data 1*. Gene names highlighted in blue were twofold upregulated and genes in pink twofold downregulated over feeding (see *Figure 2*). A majority of genes encoding outer surface proteins increased in expression throughout feeding, while having different magnitudes of expression.

The online version of this article includes the following source data for figure 4:

**Source data 1.** Transcripts per million (TPM) for all *Bb* genes across feeding timepoints.

of the DE genes would fall into these categories. We classified the genes as related to either: cell envelope, bacteriophage, cell division, DNA replication and repair, chemotaxis and motility, metabolism, transporter proteins, transcription, translation, stress response, protein degradation, or unknown (*Drecktrah et al., 2015*; see *Figure 3—source data 1*).

Of the genes that increased twofold over feeding, the clear majority each day of feeding fell into two broad categories, cell envelope (55 of 158 total genes) and proteins of unknown function (69 of 158 total genes), with fewer genes related to metabolism, chemotaxis and motility, transporters, bacteriophage, cell division, and transcription (*Figure 4A*). In contrast to the upregulated genes, genes downregulated during feeding were more evenly distributed among the functional categories – including translation, protein degradation, transcription, and metabolism – consistent with many of them being located on the chromosome.

When looking at changes across these functional categories, the overrepresentation of cell envelope proteins was striking, while not unexpected. The *Bb* outer surface is covered with lipoproteins (*Figure 4B*), and these proteins are critical determinants in *Bb* interactions with the various environments encountered during the enzootic cycle (*Kurokawa et al., 2020*). We found that more than half

(46 of 83) of annotated outer surface lipoproteins (*Dowdell et al., 2017*; *Iyer et al., 2015*) changed expression twofold over the time course (*Figure 4C*). These data suggested widespread changes may be occurring on the *Bb* outer surface during feeding.

To understand the functional implications of expression changes in a majority of outer surface lipoproteins, we also compared their relative expression. The magnitude of expression varied greatly, with *ospA*, *ospB*, *ospC*, and *bba59* being the most highly expressed outer surface protein genes. Many of the outer surface protein genes that we found had increased expression over feeding were much less abundantly expressed (*Figure 4C* and *Figure 4—source data 1*). However, even the genes that appeared to have low expression in these population level measurements could play important roles in transmission if they are highly expressed in a small number of crucial cells, such as those that ultimately escape the midgut. While bulk RNA-seq cannot distinguish what is happening at the single-cell level, our data suggest that during the bloodmeal, *Bb* are undergoing a complex outer surface transformation driven by increases in transcription of a majority of the genes encoding these lipoproteins.

## Identification of candidate tick interaction partners of *Bb* cells ex vivo

Our RNA-seq data suggested that the outer surface of *Bb* transforms over the course of feeding as *Bb* are primed for transmission to a vertebrate host. At the same time, the tick midgut environment is changing as the tick begins to digest its bloodmeal (*Sonenshine and Anderson, 2014*). Tick-*Bb* interactions are likely crucial at the beginning of the tick bloodmeal, as *Bb* adhere to the tick gut epithelium before becoming motile and migrating out of the midgut and into the salivary glands (*Dunham-Ems et al., 2009*). Some *Bb* outer surface proteins, such as BBE31 and BBA52 play key roles

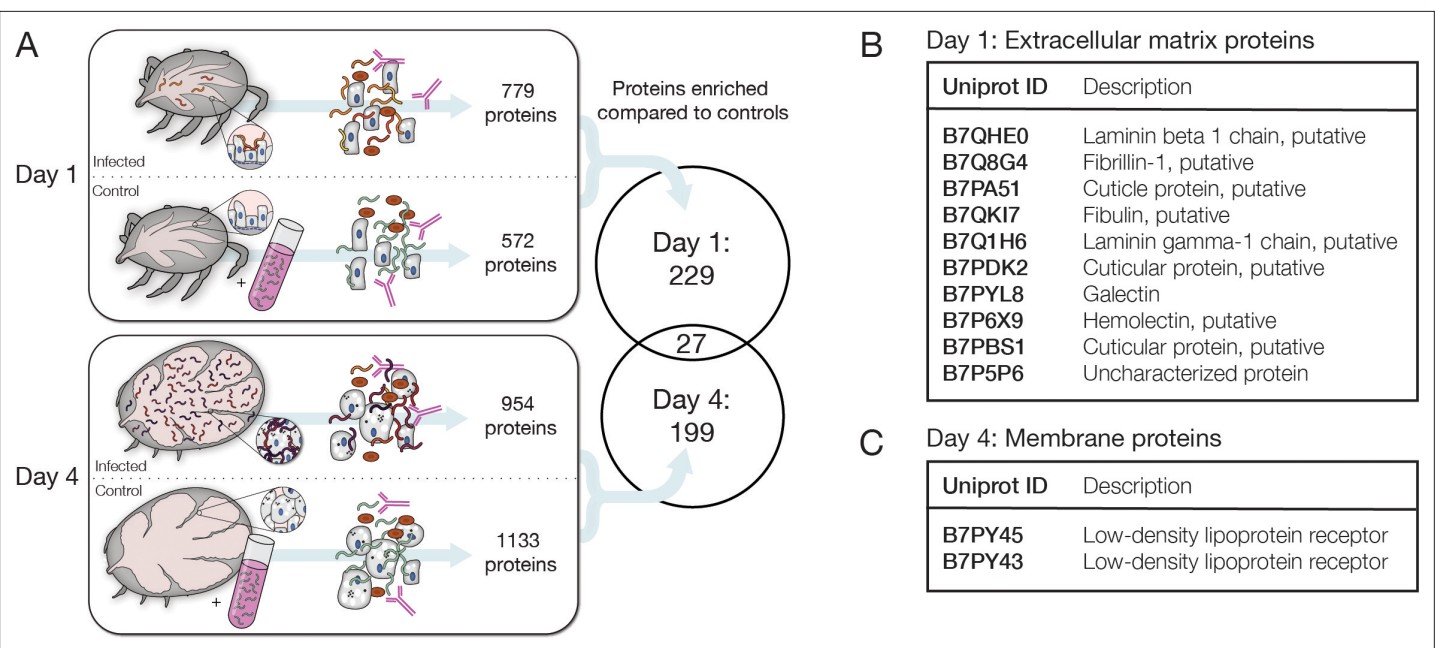

**Figure 5.** Identification of candidate tick interaction partners of *Bb* cells ex vivo. (**A**) Schematic of experiment to determine candidate tick proteins interacting with *Bb* over the course of feeding. Ticks were collected 1 day and 4 days after placement on mice. Uninfected ticks at the same time points were mixed with cultured *Bb* as controls. *Bb* was enriched with α*Bb* antibody as in RNA-seq experiments and then subjected to mass spectrometry to identify tick proteins present in the samples. Venn diagram depicts the proteins enriched in day 1 and day 4 samples over controls in at least two of three replicates, see *Figure 5—source data 1* for all proteins. Tick proteins that are enriched with *Bb* vary greatly over the course of feeding. (**B**) Tick proteins uniquely identified one day after placement that are annotated as extracellular matrix (ECM) proteins, see *Figure 5—source data 2*. (**C**) Tick proteins uniquely identified four days after attachment that are annotated as low-density lipoprotein receptors, see *Figure 5—source data 3*. ECM and membrane proteins may be good candidates for *Bb*-interacting proteins.

The online version of this article includes the following source data for figure 5:

**Source data 1.** Mass spectrometry analysis for *Bb*-enriched samples.

**Source data 2.** Annotation and GO term enrichment for tick proteins enriched on feeding day 1.

**Source data 3.** Annotation and GO term enrichment for tick proteins enriched on feeding day 4.

in pathogen migration through interactions with the tick environment (reviewed in *Kurokawa et al., 2020*). We wanted to explore the changing tick environment to identify tick proteins with which *Bb* could interact throughout the tick bloodmeal. Since our *Bb* enrichment process retained some tick material, we reasoned that tick proteins that interact with *Bb* would be present in these samples.

To outline the changes occurring in the tick during feeding and to identify candidate *Bb*-interacting tick proteins, we used mass spectrometry to survey the content of the tick material that was enriched along with the *Bb* cells we sequenced at early and late stages during feeding. We purified proteins from the α*Bb*-enriched fraction of crushed infected ticks one day after attachment and four days after attachment in triplicate. As controls for each day, we also performed the *Bb* enrichment process on lysate from uninfected ticks mixed with in vitro cultured *Bb* to help rule out proteins that were not pulled down through in vivo tick-*Bb* interactions (*Figure 5A*). When querying against the *Bb* and *I. scapularis* proteomes, we identified between 414 and 2240 protein groups per sample replicate. The vast majority of all detected proteins were from ticks (2801/2858, 98%). To identify proteins of interest, we looked for those that were detected in at least two of three replicates within infected ticks and had a mean average coverage twice that of uninfected ticks mixed with cultured *Bb*. We found 256 proteins (251 from *I. scapularis* and 5 from *Bb*) that were enriched with *Bb* from infected ticks one day after attachment and 226 proteins (220 from *I. scapularis* and 6 from *Bb*) that were enriched with *Bb* from infected ticks four days after attachment (*Figure 5—source data 1*). Of these proteins, only 27 (24 from *I. scapularis* and 3 from *Bb*) were detectable on both days, suggesting the tick proteins present upon *Bb* enrichment change dramatically over the course of feeding (*Figure 5A*). Amongst the small number of *Bb* proteins, we identified OspC in the samples from day 4 after attachment but not day 1 after attachment, confirming the expression change we saw in our RNA-seq. The distinct sets of tick proteins we identified at each timepoint suggest dramatic changes occur in the *Bb*-infected tick midgut environment during feeding that may alter the landscape of tick-*Bb* interactions.

Some of the proteins enriched with *Bb* from the tick may be good candidates for key *Bb*-interacting partners during feeding, especially if they are localized to the surface of tick cells where they may encounter *Bb*. The scarcity of both predicted and experimentally validated functions and localizations for tick proteins makes it difficult to fully assess the potential for tick protein interactions with extracellular *Bb*. Nevertheless, of the proteins found exclusively one day after attachment, 10 were categorized as extracellular matrix proteins using the PANTHER gene database (*Thomas et al., 2022*), and this category was statistically enriched (Fisher's exact test, FDR = 0.000093; *Figure 5B* and *Figure 5—source data 2*). 30 additional proteins were annotated with a cellular component as plasma membrane. Four days after attachment, we did not detect any annotated extracellular matrix proteins; however, we identified 31 proteins that are likely to be found at the membrane, including two proteins annotated as putative low-density lipoprotein receptors (*Figure 5C* and *Figure 5—source data 3*). These extracellular matrix and membrane proteins may be the most likely to directly interact with *Bb* during this timeframe and are candidates for tick proteins important in the *Bb* dissemination process. The proteins present in the changing tick environment may be key determinants of pathogen transmission as *Bb* remodels its outer surface while preparing to migrate through the tick to a new host.

## Discussion

Vector-borne pathogens must adapt to distinct environments as they are transmitted by arthropods to colonize new bloodmeal hosts. The tick-borne Lyme disease pathogen *Bb* undergoes transcriptional changes inside its vector over the course of a single bloodmeal that are important for a number of key transmission events. For example, expression changes enable survival in the feeding tick (*He et al., 2011*), facilitate transmission across several internal compartments into the bloodmeal host (*Kurokawa et al., 2020*), and prime bacteria for successful infection of the next host (*Kasumba et al., 2016*). Capturing these changes as they occur in vivo has been challenging due to the low relative abundance of *Bb* material inside of the rapidly growing, blood-filled tick. To date, many advances in our understanding of *Bb* expression during transmission from tick to host have come from tracking changes in small subsets of genes during tick feeding (*Bykowski et al., 2007*; *Gilmore et al., 2001*; *Narasimhan et al., 2002*), leveraging *Bb* culture conditions that approximate environmental changes across its lifecycle (*Ojaimi et al., 2005*; *Revel et al., 2002*; *Tokarz et al., 2004*), and defining transcriptional regulons outside of the tick (*Caimano et al., 2019*; *Caimano et al., 2015*; *Drecktrah et al., 2015*). Here, we developed an experimental RNA-seq-based strategy to more directly and

longitudinally profile gene expression for *Bb* populations isolated from ticks over the course of a transmitting bloodmeal.

Our longitudinal collection of transcriptomes for *Bb* cells isolated from ticks serves as a resource and starting point for delineating the functional determinants of *Bb* adaptation and transmission. We focused our analysis on 192 genes that changed twofold between one day after attachment and later days in feeding. While these genes were highly concordant with those found in previous studies probing transcriptional changes throughout the *Bb* enzootic cycle (*Caimano et al., 2019*; *Caimano et al., 2015*; *Drecktrah et al., 2015*; *Grassmann et al., 2023*; *Iyer et al., 2015*; *Ojaimi et al., 2003*; *Revel et al., 2002*), our transcriptome profiles revealed changes in expression of 26 genes not observed in these previous studies. These novel gene expression changes, like all of the 192 DE genes, were concentrated in genes of unknown function and those encoding outer surface proteins. Although challenging to address, efforts focused on uncovering the contribution of the genes of unknown function to the *Bb* life cycle could be groundbreaking for understanding the unique aspects of tick-borne transmission. Our data reveal more extensive expression changes for *Bb* outer surface proteins than previously appreciated, with nine additional genes encoding cell envelope proteins upregulated over feeding that were not found to change expression in the previous studies examining conditions mimicking unfed ticks versus fed ticks or in nymphs versus DMCs. The extensive changes in genes encoding outer surface proteins suggest *Bb* is actively remodeling its outer coat to navigate the dynamic tick environment during feeding, akin to wardrobe changes across seasons. These outer surface changes may play roles in the persistence of *Bb* in the tick, cell adhesion, or in immune evasion, either inside of the tick or later in the vertebrate host (*Kenedy et al., 2012*). It has long been observed that molecular interactions between pathogens and the midgut of their vectors are key determinants of transmission (*Barillas-Mury et al., 2022*). Modifications to the *Bb* surface could facilitate different tick–pathogen interactions that are critical for its physical movement through tick compartments.

More comprehensive knowledge about *Bb* outer surface proteins and their functional consequences could lead to new avenues for curtailing the spread of Lyme disease. Protective vaccines against vector-borne pathogens have often targeted surface proteins encoded by pathogens (*Kovacs-Simon et al., 2011*). While previous Lyme disease vaccine efforts effectively targeted the highly expressed *Bb* outer surface protein OspA (*Steere et al., 1998*), more targets could increase the likelihood of a successful vaccine. Our study provides a catalog of new *Bb* candidates that could be explored. In addition, our biochemical pull-downs using tick-isolated *Bb* cells as bait unearthed a preliminary list of potential tick proteins that could be involved in tick–*Bb* interactions. Blocking molecular interactions that are functionally critical for transmission could also be explored as a therapeutic strategy (*Barillas-Mury et al., 2022*; *Manning and Cantaert, 2019*).

To identify the genes changing expression during tick feeding, we used an antibody targeting *Bb* to overcome the low abundance of *Bb* in the tick. This method produced broad coverage of the transcriptome over our time course, but there are caveats. While we confirmed that the hour-long enrichment procedure does not cause significant gene expression changes in cultured *Bb*, this may still influence gene expression in our ex vivo samples. The α*Bb* antibody, which targets OspA and a number of other proteins, captured the vast majority, but not all, *Bb flaB* transcripts from inside of the tick. This loss may introduce some degree of bias. Further, a large number of day 4 reads mapping to *P. fulva* suggests that this particular antibody may enrich other bacterial species. It is also important to note that our method was unable to produce consistent expression data from *Bb* from unfed ticks, limiting the full longitudinal scope of the study and potentially overlooking key changes occurring in the first 24 hours of feeding. We predict this difficulty may be due to low numbers of *Bb* in flat nymphs (*de Silva and Fikrig, 1995*) or the stress of the prolonged nutrient-deprived period between tick feedings (*Kung et al., 2013*). The transcriptional landscape of unfed ticks may be better probed through enrichment after RNA-seq library preparation such as TBDCapSeq (*Grassmann et al., 2023*) or perhaps a combination of the two enrichment strategies. Despite these caveats, this type of antibody-based enrichment strategy is a simple and flexible technique that can probe gene expression from *Bb* from multiple timepoints in feeding ticks. With small modifications, this protocol may be able to assist in facilitating sequencing of *Bb* or other *Borrelia* species from other milieu or enrich other tick-borne bacterial species.

Our method has produced a transcriptomic resource providing critical insights into *Bb* population-level changes during the vector stage of its lifecycle, which serves as a starting point for understanding the primary drivers of tick-borne transmission. Strikingly few *Bb* cells out of the total pathogen population in ticks are ultimately transmitted to the next host during feeding (*Dunham-Ems et al., 2009*; *Rego et al., 2014*). There may be important molecular variations across *Bb* cells within the population residing in the tick that contribute to these differential outcomes including heterogeneously expressed proteins across cells within the feeding tick midgut (*Ohnishi et al., 2001*). Genes that appear unaltered or show low expression levels across the bulk population could still play an outsized role in infection for a minority of the cells, and conducting transcriptomic analyses at the single-cell level will be key. Our work provides a foundational methodology that can be leveraged to greatly improve the resolution of tick–microbe studies, which may unearth surprising mechanistic insights into the unique lifestyles of tick-borne pathogens.

# Methods

## Key resources table

| Reagent type (species) or resource | Designation | Source or reference | Identifiers | Additional information |
|---|---|---|---|---|
| Strain (*Borrelia burgdorferi*) | B31-S9 | Sourced from Dr. Patricia Rosa | | Streptomycin resistant strain |
| Strain (*Ixodes scapularis*) | *Ixodes scapularis* nymphal ticks | Sourced from Tick Lab at Oklahoma State University (OSU) (RNA-seq) and BEI Resources (mass spec) | BEI Cat# NR-44115 | Ticks were sourced as larvae; fed on mice and allowed to molt to nymphs in lab |
| Strain (*Mus musculus*) | C3H/HeJ | Jackson Laboratories | Strain#:00659 | Tick hosts were 4–6 week old female mice |
| Strain (*Borrelia burgdorferi*) | B31-A3 | Sourced from Dr. Patricia Rosa | | Used for anti-*Bb* western only |
| Strain (*Borrelia burgdorferi*) | ospA1 | Sourced from Dr. Patricia Rosa; *Battisti et al., 2008* | | ospA mutant; Used for anti-*Bb* western only |
| Strain (*Borrelia burgdorferi*) | ospA+B1 | Sourced from Dr. Patricia Rosa; *Battisti et al., 2008* | | ospA restored; Used for anti-*Bb* western only |
| Antibody | anti-*Borrelia burgdorferi* (Rabbit polyclonal) | Thermo Fisher Scientific | Invitrogen: PA1-73004; RRID: AB_1016668 | 10 mg added to immunomagnetic enrichment; IF(1:100), WB (1:10,000) |
| Antibody | anti-Rabbit HRP secondary (goat polyclonal) | Advansta | Advansta: R-05072–500; RRID: AB_10719218 | WB (1:5000) |
| Antibody | Anti-Rabbit IgG (H+L) Alexa 488 (goat polyclonal) | Thermo Fisher Scientific | Invitrogen: A-11008; RRID: AB_143165 | IF (1:100) |
| Commercial assay or kit | Dynabeads Protein G | Thermo Fisher Scientific | Invitrogen 10003D | |
| Commercial assay or kit | Zymo Direct-zol RNA Microprep Kit | Zymo Research | R2062 | With on column DNase |
| Commercial assay or kit | NEBNext Ultra II Directional RNA Library Prep Kit for Illumina | New England Biolabs | E7760L | |

*Continued on next page*

*Continued*

| Reagent type (species) or resource | Designation | Source or reference | Identifiers | Additional information |
|---|---|---|---|---|
| Commercial assay or kit | NEBNext Multiplex Oligos for Illumina Dual Index | New England Biolabs | E7600S | |
| Commercial assay or kit | Kapa HiFi Real-Time Library Amplification Kit | Roche | Kapa KK2702 | |
| Commercial assay or kit | Cas9 | New England Biolabs | M0386S | |
| Commercial assay or kit | Taqman Universal PCR Master Mix | Thermo Fisher Scientific | Applied Biosystems 4304437 | |
| Commercial assay or kit | PowerUp SYBR Green Master Mix | Thermo Fisher Scientific | Applied Biosystems A25741 | |
| Sequence-based reagent | crRNAs targeting tick, mouse, *Bb* rRNA | Dynerman et al.; *Ring et al., 2022* | | See *Supplementary file 1* |
| Sequence-based reagent | flaB F | *Jewett et al., 2007* | qPCR primers | 5'- TCTTTTCTCTGGTGAGGGAGCT |
| Sequence-based reagent | flaB R | *Jewett et al., 2007* | qPCR primers | 5'-TCCTTCCTGTTGAACACCCTCT |
| Sequence-based reagent | flaB probe | *Jewett et al., 2007* | qPCR probe | /56-FAM/AAACTGCTCAGGCTGCACCGGTTC/36-TAMSp |
| Sequence-based reagent | gapdh F | This paper | qPCR primer | 5'-TTCATTGGAGACACCCACAG |
| Sequence-based reagent | gapdh R | This paper | qPCR primer | 5'-CGTTGTCGTACCACGAGATAA |
| Chemical compound | Propidium iodide | Thermo Fisher Scientific | Invitrogen P3566 | |
| Chemical compound | TRIzol | Thermo Fisher Scientific | Invitrogen 15596018 | |
| Software, algorithm | DASHit | *Dynerman et al., 2020* | | http://dashit.czbiohub.org/ |
| Software, algorithm | Salmon | *Patro et al., 2017* | RRID:SCR_017036 | |
| Software, algorithm | STAR | *Dobin et al., 2013* | RRID:SCR_004463 | |
| Software, algorithm | Bowtie2 | *Langmead and Salzberg, 2012* | RRID:SCR_016368 | |
| Software, algorithm | DESeq2 | *Love et al., 2014* | RRID:SCR_015687 | |
| Software, algorithm | GraphPad Prism v9.5.1 | GraphPad Software | RRID:SCR_002798 | |
| Software, algorithm | PEAKS Online Xpro 1.6 | Bioinformatics Solutions Inc. | RRID:SCR_022841 | |
| Other | BSK II | Rosa Lab Recipe | | Media used to grow Bb; https://www.niaid.nih.gov/sites/default/files/lzp_recipes.pdf |

### *B. burgdorferi* culture

*Bb* strain B31-S9 (*Rego et al., 2011*) was provided by Dr. Patricia Rosa (NIAID, NIH, RML) and cultured in BSK II media at 35 °C, 2.5% $CO_2$. B31-S9 was used for all RNA-seq and *Bb* enrichment experiments. Wildtype *Bb* strain B31-A3, ospA1-mutants (ospA1) and ospA-restored *Bb* (ospA$^+$B1) *Battisti et al., 2008* used in αBb western blot were also provided by Dr. Rosa.

## Tick feeding experiments

*I. scapularis* larvae were purchased from the Tick Lab at Oklahoma State University (OSU) for RNA-seq experiments or provided by BEI Resources, a division of the Center for Disease Control, for mass spectrometry experiments. Before and after feeding, ticks were maintained in glass jars with a relative humidity of 95% (saturated solution of potassium nitrate) in a sealed incubator at 22 °C with a light cycle of 16 hr/8 hr (light/dark). Animal experiments were conducted in accordance with the approval of the Institutional Animal Care and Use Committee (IACUC) at UCSF, Project Number AN183452. Ticks were fed on young (4–6 week-old) female C3H/HeJ mice acquired from Jackson Laboratories. Mice were anesthetized with ketamine/xylazine before placement of ≤100 larval or ≤30 nymphal ticks. Replete larval ticks were placed in the incubator to molt before being used as nymphs in experiments. Nymphal ticks were either pulled off isoflurane anesthetized mice at various times during feeding (1–3 days after placement) or allowed to feed to repletion and collected from mouse cages (4 days after placement).

## Western blot with α*Bb* antibody

To determine whether the α*Bb* antibody targeted ospA, wildtype *Bb* (B31-A3), ospA1-mutants (ospA1) and ospA-restored *Bb* (ospA$^+$B1) (*Battisti et al., 2008*) were cultured to approximately 5x10$^7$ *Bb*/mL. 3 mLs of culture were centrifuged for 7 min at 8000 x *g*, washing twice with PBS. Pelleted cells were lysed in 50 µL of water, and 25 µg of protein per sample were mixed with 5 X loading dye (0.25% Bromophenol Blue, 50% Glycerol, 10% Sodium Dodecyl Sulfate, 0.25 M Tris-Cl pH 6.8, 10% B-Mercaptoethanol), run on a Mini-PROTEAN TGX 4–15% gel (Bio-Rad, Hercules, CA), and transferred using the Trans-Blot Turbo Transfer System (Bio-Rad). After transfer, the blot was blocked for 30 minutes at 4 °C in TBST (Tris buffered saline with 0.1% tween) with 5% milk, then treated with α*Bb* antibody (Invitrogen, Waltham, MA: PA1-73004; RRID: AB_1016668) diluted 1:10,000 for 1 hr at room temperature, followed by anti-rabbit HRP secondary antibody (Advansta, San Jose, CA: R-05072–500; RRID: AB_10719218) diluted 1:5000 for 45 min at room temperature with three short PBST washes between each step. Blots were exposed using Clarity Western ECL Substrate (Bio-Rad) and imaged using the Azure C400 imaging system (Azure Biosystems, Dublin, CA). This experiment was repeated three times.

## Enrichment of *Bb* from feeding ticks

To sequence RNA from *Bb* inside of feeding ticks, *Bb* were enriched to increase the ratio of *Bb* to tick material. Larval ticks were fed to repletion on three mice that were infected with *Bb* through intraperitoneal and subcutaneous injection with 10$^4$ total *Bb*. Approximately 5 months later, the molted nymphal ticks were fed on eight mice, which were housed individually during the feeding. We estimated that 83% of the ticks were infected with *Bb* by crushing 12 unfed nymphs in BSK II media and checking for viable *Bb* days later. Ticks were pulled from all mice and pooled into four biological replicates 1 day after placement (14 ticks per replicate), 2 days after placement (12 ticks per replicate), and 3 days after placement (6 ticks per replicate) and collected from cages 4 days after placement (7 ticks per replicate). Shortly after collection, ticks were washed with water and placed in a 2 mL glass dounce grinder (Kimble, DWK Life Sciences, Millville, NJ) in 500 µL of phosphate-buffered saline (PBS). Ticks were homogenized first with the large clearance pestle and then the small clearance pestle. The homogenate was transferred to a 1.5 mL Eppendorf tube and 500 µL of PBS was added to total 1 mL. At this stage, 50 µL of homogenate was removed as an input sample and mixed with 500 µL of TRIzol (Invitrogen) for RNA extraction. Two µL of α*Bb* antibody (Invitrogen: PA1-73004; RRID: AB_1016668) was added to the homogenate, which was then placed on a nutator at 4 °C for 30 min. During incubation, 50 µL of Dynabeads Protein G (Invitrogen) per sample were washed twice in PBS. After incubation with the antibody, the homogenate and antibody mixture were added to the beads. This mixture was placed on a nutator at 4 °C for 30 min. Tubes were then placed on a magnet to secure beads, and the homogenate was removed and saved to create depleted samples. The depleted homogenate was centrifuged at 8000 x *g* for 7 min, 900 µL of supernatant was removed, and 500 µL of TRIzol was added to the pellet to create depleted samples. The beads were washed twice with 1 mL of PBS, resuspending the beads each time. The second wash was removed and 500 µL of TRIzol was added to the beads to create enriched samples. RNA was extracted from all input, enriched, and depleted samples using the Zymo Direct-zol RNA Microprep Kit with on-column

DNase treatment (Zymo Research, Irvine, CA). The step-by-step *Bb* enrichment protocol is available at: https://dx.doi.org/10.17504/protocols.io.36wgqjrbovk5/v1.

## Enrichment of *Bb* from culture

To test whether the *Bb* enrichment process altered gene expression levels, we performed the enrichment protocol on cultured *Bb*. Tubes of *Bb* in BSK II media were grown to $9 \times 10^4$ *Bb*/mL at 35 °C. 1 mL of culture was spun down at 8000 x *g* for 7 min, media was removed, *Bb* were washed in 1 mL of PBS and spun again. Pelleted *Bb* were resuspended in 1 mL of fresh PBS. These samples were used as starting homogenate for the *Bb* enrichment protocol and input, enriched, and depleted fractions were collected as above. RNA-seq libraries from these samples were prepared and sequenced as below.

## RNA-seq library preparation and sequencing

To make RNA-seq libraries from enriched *Bb* RNA, 50 ng of total RNA was used as input into the NEBNext Ultra II Directional RNA Library Prep Kit for Illumina (New England Biolabs, Ipswich, MA). Libraries were prepared following the manufacturer's protocol for use of the kit with purified mRNA or rRNA-depleted RNA, despite starting with total RNA. Libraries were barcoded using NEBNext Multiplex Oligos for Illumina Dual Index (New England Biolabs).

To deplete reads from the libraries that were from highly expressed tick, *Bb*, or mouse rRNA sequences, we used Depletion of Abundant Sequences by Hybridization (DASH *Gu et al., 2016*), which targets Cas9 to unwanted reads in RNA-seq libraries using custom dual-guide RNAs (dgRNAs). The dgRNAs targeted short sequences within tick rRNA, mouse rRNA, and *Bb* rRNA that were designed using DASHit software (*Dynerman et al., 2020*). We ordered crRNA oligos (*Supplementary file 1*) that targeted these sequences (*Ring et al., 2022*). To transcribe the dgRNAs, we followed the protocol for In Vitro Transcription for dgRNA V2 (*Lyden et al., 2019b*) as follows. Both tracrRNA and pooled crRNA DNA templates were annealed to equimolar amounts of T7 primer by heating to 95 °C for 2 min and slowly cooling to room temperature. Annealed templates were used in 1 mL in vitro transcription reactions with: 120 µL 10X T7 buffer (400 mM Tris pH 7.9, 200 mM MgCl2, 50 mM DTT, 20 mM spermidine (Sigma-Aldrich, St. Louis, MO)), 100 µL of T7 enzyme (custom prepped enzyme from E. Crawford, diluted 1:100 in T7 buffer, final concentration: 100 µg/mL), 300 µL NTPs (25 mM each, Thermo Fisher Scientific, Waltham, MA), 4 µg of annealed crRNA template or 8 µg of annealed tracrRNA template, and water to 1 mL. In vitro transcription was performed for 2 hr at 37 °C. Reactions were purified twice with the Zymo RNA Clean & Concentrator-5 Kit (Zymo Research). To form the dgRNA complex for DASH, crRNA and tracrRNAs were diluted to 80 µM, mixed in equimolar amounts and annealed by heating to 95 °C for 30 s and cooling slowly to room temperature.

After transcription of dgRNAs, we performed DASH Protocol Version 4 (*Lyden et al., 2019a*) on each library individually as follows. Cas9 and transcribed dgRNAs were prepped by mixing: 2.5 µL 10 X Cas9 buffer, 5 µL 20 µM Cas9 (New England Biolabs), and 5 µL of 40 µM transcribed dgRNAs. The mixture was incubated at 37 °C for 5 min before 7.5 µL of RNA-seq library (2.8 nM) was added. The mixture was incubated at 37 °C for 1 hr and then purified with Zymo DNA Clean & Concentrator-5 (Zymo Research) following the PCR product protocol and eluting DNA into 10.5 µL of water. During cleanup, Cas9 was again mixed with buffer and dgRNAs and incubated at 37 °C for 5 min. Following cleanup, eluted DNA was added to the second Cas9-dgRNA mixture and incubated at 37 °C for 1 hr for a second time. Then, 1 µL of proteinase K (New England Biolabs) was added, and the mixture was incubated at 50 °C for 15 min. The libraries were then purified with 0.9 X volume of sparQ PureMag Beads (QuantaBio, Beverly, MA) following the standard protocol, eluting in 24 µL of water. rRNA-depleted RNA-seq libraries were then amplified in a Bio-Rad CFX96 using the Kapa HiFi Real-Time Amplification Kit (Roche, Basel, Switzerland) in a 50 µL reaction with 25 µL master mix, 23 µL of the DASHed library pool, and 2 µL of a 25 µM mix of Illumina P5 (5'-AATGATACGGCGACCACCGAGATCT) and P7 (5'- CAAGCAGAAGACGGCATACGAGAT) primers. The qPCR program for amplification was as follows: 98 °C for 45 sec (1 cycle), (98 °C for 15 s, 63 °C for 30 s, 72 °C for 45 s, plate read, 72 °C for 20 s) for 10 cycles (day 3 and day 4 samples) or 11 cycles (day 1 and day 2 samples). The libraries were removed from cycling conditions before leaving the exponential phase of amplification and then purified with 0.9 X volume of sparQ PureMag Beads according to the standard protocol.

Following DASH, RNA-seq libraries were sequenced on an Illumina NovaSeq S2 (2 lanes) with paired-end 100 base pair reads. Libraries from in vitro cultured control experiment were sequenced

on an Illumina NextSeq with paired-end 75 base pair reads. FASTQ files and raw *Bb* read counts for in vitro control experiment (GSE217146) and ex vivo experiment (GSE216261) have been deposited in NCBI's Gene Expression Omnibus (*Edgar et al., 2002*) under SuperSeries accession number GSE217236.

## RNA-seq data analysis
### DASH
To measure the success of our rRNA depletion through DASH, we used DASHit software (*Dynerman et al., 2020*) to determine the percentage of reads that would be DASHable by our guide RNAs (*Ring et al., 2022*). For pre-DASH data, we sequenced the input of each of our RNA-seq libraries before performing DASH on a MiSeq V2 Micro (Illumina, San Diego, CA). We tested DASHability on a random subset of 200,000 paired-end reads chosen by seqtk v1.3 (RRID:SCR_018927) from each pre- and post-DASH library. Paired t test comparing DASHable reads before and after DASH was performed using GraphPad Prism v9.5.1 (GraphPad Software, San Diego, CA).

## Differential expression analysis
To map our RNA-seq data to *Bb*, we wanted to optimize for mapping reads that came from the many paralogous gene families found across the plasmids of the genome. We used the pseudoalignment tool Salmon v1.2.1 (*Patro et al., 2017*), which is used to accurately map reads coming from different isoforms of the same gene, for this reason. While using Salmon to map to gene sequences may improve mapping to paralogous genes, it may also have a tradeoff of reduced mapping of reads that fall on the ends of genes that reside in operons. Nevertheless, all samples should be similarly affected, and any undercounting should not change differential expression results. Reads were first trimmed of bases with quality scores less than 20 using Cutadapt (*Martin, 2011*) via Trim Galore v0.6.5 (RRID:SCR_011847). Reads were mapped to *Bb* gene sequences (for all protein coding and pseudogenes in the Genbank feature table) as a reference transcriptome from NCBI Genbank GCA_000008685.2 ASM868v2 (with plasmids lp5, cp9, and lp56 removed as they are not present in B31-S9) using Salmon with the following parameters: `--validateMappings --seqBias --gcBias`. Before mapping, the transcriptome was indexed using the Salmon index command with the whole genome as decoys and the parameter `--keepDuplicates` to keep all duplicate genes.

Read counts from Salmon were used as input into DESeq2 v1.24.0 (*Love et al., 2014*) for differential expression analysis in R version 3.6.1. DESeq2 function PlotPCA() was used to create a PCA plot from read counts after running the varianceStabilizingTransformation() function. For differential expression analysis between days, a DESeq object was created from count data using the DESeq() function. The lfcShrink() function with the apeglm method (*Zhu et al., 2019*) was used to calculate fold changes between days. DESeq2 uses a Benjamini-Hochberg multiple testing correction, and we focused the majority of our analysis on genes that had an adjusted p-value <0.05 and used an additional cutoff requiring genes to change twofold between conditions. Code used for differential expression analysis is available at: https://github.com/annesapiro/Bb-tick-feeding (copy archived at *Sapiro et al., 2023*).

## Mapping to other species
To determine the source of non-*Bb* reads in our RNA-seq libraries, trimmed reads were mapped to tick and mouse genomes using STAR v2.7.3a (*Dobin et al., 2013*). The *I. scapularis* ISE6 genome (RefSeq assembly GCF_002892825.2, ISE6_asm2.2_dedeplicated) (*Miller et al., 2018*) was indexed using STAR run mode genomeGenerate with option `--genomeChrBinNbits` 18. The *Mus musculus* genome GRCm39 (RefSeq assembly GCF_000001635.27) was indexed using STAR run mode genomeGenerate with basic options. Reads were mapped using STAR to each genome using basic options. The percentage of reads that mapped to these genomes was determined by adding the percentage of uniquely mapped reads, reads mapped to multiple loci, and reads mapped to too many loci. To identify the potential source of reads that did not map to tick, mouse, or *Bb* in day 4 samples, one million reads from day 1 and day 4 libraries were used as input into CZ ID (*Kalantar et al., 2020*), which determined that a large number of reads mapped to bacterial species *Pseudomonas fulva*. Full RNA-seq libraries were then mapped to the *P. fulva* genome (NCBI GenBank GCF_001186195.1 ASM118619v1), using the standard options of Bowtie2 (*Langmead and Salzberg, 2012*) to calculate the overall alignment rate.

## Comparisons to other studies

Genes identified in previous studies were compared to time course expression changes. Here, we considered RpoS-regulated genes as those found in *Grassmann et al., 2023* that were upregulated by RpoS in both fed nymphs and DMCs and those upregulated by RpoS only during tick transmission (Grassmann et al. Supplemental Tables 5 and 6). RpoS did not suppress the expression of any genes in fed nymphs in the study. Genes up- and down-regulated by RpoS in DMCs only (Grassmann et al. Supplemental Tables 7 and 8) are noted in *Figure 2—source data 2* and *Figure 3—source data 1* for reference along with genes found to be regulated by RpoS in DMCs in *Caimano et al., 2019* (Tables 2 and 3), which were used for RpoS comparisons in previous versions of this study. Rrp1 up- and down-regulated genes were those identified in vitro in *Caimano et al., 2015*, Table S2. Rel$_{Bbu}$ up- and down-regulated genes were examined by *Drecktrah et al., 2015* in three different in vitro conditions: starvation (Tables S6 and S9), recovery (Tables S7 and S10), and stationary phase (Tables S5 and S8). For simplicity, we considered genes as Rel$_{Bbu}$-regulated if they were up- or down-regulated in one or more of these conditions (*Figure 2—source data 2* and *Figure 3—source data 1*). One gene was regulated in opposing directions across conditions and is noted in our tables as 'both' and was excluded from the comparison analysis. Genes changing between 'unfed tick' and 'fed tick' culture conditions in *Revel et al., 2002* were those found in Table 3. Genes from *Ojaimi et al., 2003* Table 4 with increased expression in vitro at 35 °C relative to 25 °C were considered higher in 'fed tick' while those in *Ojaimi et al., 2003* Table 5 with increased expression at 25 °C relative to 35 °C were considered higher in 'unfed tick'. Genes more highly expressed in nymphs than DMCs from *Iyer et al., 2015* were found in Table S4, and genes more highly expressed in DMCs than nymphs were found in Table S8. Genes differentially expressed between nymphs and DMCs in *Grassmann et al., 2023* were determined from the DESeq2 comparison between WT DMC vs Fed Nymphs found in Supplemental Table 3, in accordance with author cutoffs of at least a threefold difference and q-value<0.05. As many of these studies used different strains of *Bb* and different genome annotations, some genes were not examined here as they were not present in the B31-S9 strain used.

## Gene classification

To classify genes into functional groups, functional categories were sourced from *Drecktrah et al., 2015* where available. Other gene functions were sourced from *Fraser et al., 1997*. Genes found within the co-transcribed 'late' bacteriophage operon (*Zhang and Marconi, 2005*) were considered 'bacteriophage' even if their function is unknown. Outer surface proteins were those found in *Dowdell et al., 2017* plus additional outer surface proteins listed in *Iyer et al., 2015* that were also found in *Dowdell et al., 2017* Supporting Table S2 categories SpII and SpI as evidence of outer surface localization. Outer surface and periplasmic lipoproteins were classified as 'cell envelope' in the absence of other classifications. Gene family information from *Casjens et al., 2000* was considered to aid in classification. *Figure 3—source data 1* contains the classification source for each gene.

## RT-qPCR measuring *Bb* enrichment

To test the efficacy of the *Bb* enrichment protocol, RT-qPCR was used to quantify *Bb flaB* and *I. scapularis gapdh* transcript levels in enriched and depleted fractions. cDNA was synthesized from 8 µL of RNA extracted from *Bb* enrichment samples and their matched depleted samples from day 2, day 3, and day 4 post-attachment using the qScript cDNA Synthesis Kit (Quantabio). cDNA was diluted 2 X before use in qPCR. To measure *flaB* copies, standards of known concentration were created from purified PCR products. These standards were made from PCR with primers with the following sequences: 5'-CACATATTCAGATGCAGACAGAGGTTCTA and 5'-GAAGGTGCTGTAGCAG GTGCTGGCTGT. A dilution series with 10-fold dilutions between $10^6$ copies and $10^1$ copies of this PCR template was run alongside enriched and depleted samples. qPCR was performed using Taqman Universal PCR Master Mix (Applied Biosystems, Waltham, MA). The primers used to amplify *flaB* were: 5'- TCTTTTCTCTGGTGAGGGAGCT and 5'-TCCTTCCTGTTGAACACCCTCT (used at 900 nM) and the probe was /56-FAM/AAACTGCTCAGGCTGCACCGGTTC/36-TAMSp (used at 250 nM) (*Jewett et al., 2007*). For tick *gapdh* RT-qPCR, the cDNA samples were diluted an additional 2 X. Standards of known concentration were created using the qPCR primer sequences: 5'-TTCATTGGAGACACCC ACAG and 5'-CGTTGTCGTACCACGAGATAA (used at 900 nM). qPCR was performed using PowerUp SYBR Green Master Mix (Applied Biosystems). For both *flaB* and *gapdh*, the number of copies in each

sample was calculated based on the standards of known concentration. Three technical replicates were averaged from each of four biological replicates at each time point tested. We totaled the number of copies in each matched enriched and depleted fraction to calculate the percentage of *flaB* or *gapdh* that was found in either sample. All qPCR was performed on the QuantStudio3 Real-Time PCR System (Applied Biosystems). Paired t tests were performed using GraphPad Prism v9.5.1.

## Immunofluorescence microscopy

To test whether the α*Bb* antibody recognized *Bb* inside of the tick, ticks at each day of feeding were crushed in 50 μL of PBS. Ten μL of lysate was spotted onto slides and allowed to air dry before slides were heated briefly three times over a flame. Heat fixed slides were then treated with acetone for 1 hr. Slides were incubated with α*Bb* primary antibody (1:100 diluted in PBS +0.75% BSA) for 30 min at 37 °C in a humid chamber. A control without primary antibody was also used for each day. Slides were washed once in PBS for 15 min at room temperature, then rinsed in distilled water and air dried. Anti-rabbit IgG Alexa 488 (Invitrogen: A-11008; RRID: AB_143165) diluted 1:100 in PBS +0.75% BSA was added for 30 min at 37 °C in a humid slide chamber. Slides were washed in PBS for 15 min at room temperature three times, adding 1:100 Propidium Iodide (Invitrogen) during the second wash. Slides were then rinsed with distilled water and air dried before the addition of mounting media (Fluoromount-G, SouthernBiotech, Birmingham, AL) and cover slips. Fluorescence imaging was performed on a Nikon Ti2 inverted microscope for widefield epifluorescence using a 100 X/1.40 objective. Images were captured with NIS-Elements AR View 5.20 and then processed with ImageJ software (*Schneider et al., 2012*). No strong florescence signal was observed on the control slides without primary antibody.

## Mass spectrometry of *Bb*-enriched samples

To identify which tick proteins were found in samples after *Bb* enrichment across feeding, both uninfected and infected ticks were fed on mice. Three biological replicates of uninfected ticks one day after placement (11 ticks per replicate), infected ticks 1 day after placement (27 ticks per replicate), uninfected ticks 4 days after placement (8 ticks per replicate), and infected ticks four days after placement (16 ticks per replicate) were collected. Before α*Bb* enrichment, the uninfected tick samples were mixed with *Bb* grown in culture that was washed with PBS ($3x10^4$ *Bb* one day after placement and $3x10^6$ *Bb* 4 days after placement) and mixed lysates were rotated at room temperature for 30 min. Infected tick samples underwent the *Bb* enrichment process immediately. The enrichment process followed the same protocol used for RNA-seq. Sample volumes were increased to 1 mL as needed, and then 2 μL of α*Bb* antibody (Invitrogen: PA1-73004; RRID: AB_1016668) was added, and samples were rotated at 4 °C for 30 min. Fifty μL of Dynabeads Protein G per sample were washed in PBS during this incubation and added to the lysates, which were rotated at 4 °C for 30 min. The beads were washed twice with 1 mL of PBS, and then placed into 50 μL of lysis buffer (iST LYSE, PreOmics, Martinsried, Bayern, Germany). Samples were boiled at 95 °C for 5 min, and lysates were removed from beads and frozen for mass spectrometry preparation.

For mass spectrometry, a nanoElute was attached in line to a timsTOF Pro equipped with a CaptiveSpray Source (Bruker, Billerica, MA). Chromatography was conducted at 40 °C through a 25 cm reversed-phase C18 column (PepSep) at a constant flowrate of 0.5 μL min−1. Mobile phase A was 98/2/0.1% water/MeCN/formic acid (v/v/v) and phase B was MeCN with 0.1% formic acid (v/v). During a 108 min method, peptides were separated by a 3-step linear gradient (5%–30% B over 90 min, 30% to 35% B over 10 min, 35% to 95% B over 4 min) followed by a 4 min isocratic flush at 95% for 4 min before washing and a return to low organic conditions. Experiments were run as data-dependent acquisitions with ion mobility activated in PASEF mode. MS and MS/MS spectra were collected with m/z 100–1700 and ions with z = +1 were excluded.

Raw data files were searched using PEAKS Online Xpro 1.6 (Bioinformatics Solutions Inc). The precursor mass error tolerance and fragment mass error tolerance were set to 20 ppm and 0.03 respectively. The trypsin digest mode was set to semi-specific and missed cleavages was set to 2. The *I. scapularis* reference proteome (Proteome ID UP000001555, taxon 6945) and *Bb* reference proteome (Proteome ID UP000001807, strain ATCC 35210/B31) were downloaded from Uniprot, totaling 21,774 entries. The *I. scapularis* proteome was the primary search reference and the *Bb* was

used as a secondary to identify any bacterial proteins present. Carbamidomethylation was selected as a fixed modification. Oxidation (M) and Deamidation (NQ) was selected as a variable modification.

Experiments were performed in biological triplicate, with samples being a single run on the instrument. Proteins present in a database search ($-10 \log(\text{p-value}) \geq 20$, 1% peptide and protein FDR) were subjected to the following filtration process. Proteins were filtered to include only those found in two out of three biological replicates, within each respective day (one or four days after placement). The mean area of proteins found in uninfected and infected samples was calculated. Proteins with missing values (i.e. not identified in a sample) were set to 1. The ratio of mean area for each protein was calculated as infected/uninfected, and enriched proteins were identified by having an infected to uninfected ratio greater than 2 within their respective feeding day (1 or 4 days after attachment).

To classify the identified proteins into functional groups, a PANTHER Overrepresentation Test (released 07/12/2022) was used (*Mi et al., 2019*) with PANTHER version 17.0 (released 02/22/2022) (*Thomas et al., 2022*). All *I. scapularis* genes in the database were used as the reference list and all proteins enriched on each day were analyzed for their PANTHER Protein Class. The test type used was Fisher's Exact, calculating a false discovery rate as the correction.

Raw data files and searched datasets are available on the Mass Spectrometry Interactive Virtual Environment (MassIVE), a full member of the Proteome Xchange consortium under the identifier: MSV000090560.

## Acknowledgements

We are grateful to all members of the Chou lab for their feedback throughout the project, specifically to Fauna Yarza and Patrick Rockefeller Grimes for assistance with tick feeding and Ethel Enoex-Godonoo for administrative assistance. We thank Amy Lyden and Emily Crawford for assistance with and reagents for DASH along with Olga Botvinnik, Michelle Tan, and The Chan Zuckerberg Biohub sequencing team for their input and sequencing assistance. We thank Patricia Rosa, Jenny Wachter, Scott Samuels, and Meghan Lybecker for their feedback on the project. We also thank William Hatleberg for assistance with figure schematics.

## Additional information

### Competing interests

Seemay Chou: Seemay Chou is president and CEO of Arcadia Biosciences. The other authors declare that no competing interests exist.

### Funding

| Funder | Grant reference number | Author |
|---|---|---|
| Life Sciences Research Foundation | | Anne L Sapiro |
| Arnold and Mabel Beckman Foundation | | Balyn W Zaro |
| Montana INBRE | P20GM103474 | Margie Kinnersley Patrick R Secor |
| Chan Zuckerberg Initiative | | Seemay Chou |
| Pew Charitable Trusts | | Seemay Chou |

The funders had no role in study design, data collection and interpretation, or the decision to submit the work for publication.

### Author contributions

Anne L Sapiro, Conceptualization, Data curation, Formal analysis, Funding acquisition, Validation, Investigation, Visualization, Methodology, Writing - original draft, Project administration, Writing – review and editing; Beth M Hayes, Conceptualization, Supervision, Funding acquisition, Validation, Investigation, Project administration, Writing – review and editing; Regan F Volk, Data curation,

Formal analysis, Investigation, Writing – review and editing; Jenny Y Zhang, Investigation, Methodology, Writing – review and editing; Diane M Brooks, Formal analysis, Investigation, Visualization, Methodology, Writing – review and editing; Calla Martyn, Formal analysis, Investigation, Methodology, Writing – review and editing; Atanas Radkov, Investigation, Writing – review and editing; Ziyi Zhao, Formal analysis, Investigation, Visualization, Writing – review and editing; Margie Kinnersley, Conceptualization, Formal analysis, Funding acquisition, Investigation, Methodology, Writing – review and editing; Patrick R Secor, Balyn W Zaro, Seemay Chou, Conceptualization, Supervision, Funding acquisition, Writing – review and editing

### Author ORCIDs
Anne L Sapiro (ID) http://orcid.org/0000-0002-6612-8272
Beth M Hayes (ID) http://orcid.org/0000-0001-6633-751X
Regan F Volk (ID) http://orcid.org/0000-0001-6748-719X
Jenny Y Zhang (ID) http://orcid.org/0000-0003-1352-8018
Patrick R Secor (ID) http://orcid.org/0000-0001-7123-3037
Balyn W Zaro (ID) http://orcid.org/0000-0002-8938-9889

### Ethics
Animal experiments were conducted in accordance with the guidelines and approval of the Institutional Animal Care and Use Committee (IACUC) at UCSF, Project Number AN183452.

Reviewer #1 (Public Review): https://doi.org/10.7554/eLife.86636.3.sa1
Reviewer #2 (Public Review): https://doi.org/10.7554/eLife.86636.3.sa2
Author Response: https://doi.org/10.7554/eLife.86636.3.sa3

---

## Additional files

### Supplementary files
- Supplementary file 1. crRNAs targeting tick, mouse, and *Bb* rRNA sequences used in DASH.
- MDAR checklist

### Data availability
For sequencing data, FASTQ files and raw *Bb* read counts for in vitro control experiment (GSE217146) and ex vivo experiment (GSE216261) have been deposited in NCBI's GEO database under SuperSeries accession number GSE217236. For mass spectrometry data, raw data files and searched datasets are available on the Mass Spectrometry Interactive Virtual Environment (MassIVE) under the identifier: MSV000090560. Code used for data analysis is available at on GitHub (copy archived at *Sapiro et al., 2023*). Source data files contain the numerical data used to generate the figures.

The following datasets were generated:

| Author(s) | Year | Dataset title | Dataset URL | Database and Identifier |
|---|---|---|---|---|
| Sapiro AL, Hayes BM, Chou S | 2022 | Longitudinal map of transcriptome changes in the Lyme pathogen Borrelia burgdorferi during tick-borne transmission | https://www.ncbi.nlm.nih.gov/geo/query/acc.cgi?acc=GSE217236 | NCBI Gene Expression Omnibus, GSE217236 |
| Volk RF, Zaro BW | 2022 | Tick proteins interacting with borrelia burgdorferi Sapiro et al 2022 | https://massive.ucsd.edu/ProteoSAFe/dataset.jsp?task=fcfb24f7b86a4beeb95a6e6e614ff8e7 | MassIVE, MSV000090560 |

---

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
