## [Editor Report · eLife assessment]

In this Tools and Resources article, the authors overcome the challenge of low *Borrelia burgdorferi* numbers during infection for analyses such as RNA-sequencing or mass spectrometry. They do so by physically enriching for spirochetes, which is **important**, as it provides technical advances for the study of global transcriptomic changes of *B. burgdorferi* during tick feeding, helping to build on the knowledge already collected by the field. The evidence presented is **compelling**, and the strategy described here could benefit researchers in the field and possibly also support broader applications.

---

## [Referee Report · Reviewer #1 (Public Review)]

In their research article, Sapiro et al. overcome the technical burden of low B. burgdorferi numbers during infection by physically enriching for spirochetes prior to RNA-sequencing/mass spectrometry. This technology, which has potential broad applications, was applied to B. burgdorferi-infected ticks, generating datasets for future studies.

Sapiro et al. addressed many of the reviewers' comments including the addition of experimental details, comparisons to other studies and some caveats to their approach. The manuscript has been significantly improved and I appreciate the efforts to address our critiques. There are a few remaining comments that the authors should consider before creating the final Version of Record.

The authors sought to develop technology for a transcriptomic analysis of B. burgdorferi directly from infected ticks. The methodology has exciting implications to better understand pathogen RNA profiles during specific infection timepoints, even beyond the Lyme spirochete. The authors demonstrate successful sequencing of the B. burgdorferi transcriptome from ticks and perform mass spectrometry to identify possible tick proteins that interact with B. burgdorferi. This technology and first dataset will be useful for the field. The study is limited in that no transcripts/proteins are followed-up by additional experiments and no biological interactions/infectious-processes are investigated.

Remaining critiques:

Experimental data regarding the sensitivity of this approach are missing. What is the limit of detection for this protocol? While the authors have stated that they were unable to sequence B. burgdorferi from unfed nymphs, the number of bacteria needed for antibody enrichment are not tested. The starting CFU in their infected nymphal ticks was also not reported (the authors only report reisolation data from 12 ticks). Page 18, line 458 the authors claim their approach "captured the vast majority" of Bb inside of the tick. Data are missing to demonstrate this. Understanding the limits of this approach will be critical for future applications, especially when using B. burgdorferi infected material with low bacterial burden.

The authors should clarify the term "genes" in the abstract and throughout the manuscript. I think they actually mean "open reading frames" or "annotated mRNAs".

More information regarding the efficacy of RNA-seq coverage is still warranted and lacking from the results, especially on page 6. The authors skip right to differential expression analysis without fully examining sequencing effectiveness. This is especially important given their development of a new technique. What was the numbers of detected genes for each sample? How is this affected by bacterial burden of the sample? What is the distribution of reads among tRNAs, mRNAs, UTRs, and sRNAs? How reproducible is the coverage for one gene across replicates? A few browser images of RNA-seq data (ex. of BAM files) across different genes would be useful to visualize the read coverage per gene.

Downregulated genes are largely ignored and should be commented on further.

Page 11, line 258-260: authors state Rpos, Rrp1, and RelBbu are the "three main Bb regulatory programs active in the tick." Yes, these three regulons have been well studied but there could be other uncharacterized regulatory programs. Please consider changing the language.

---

## [Referee Report · Reviewer #2 (Public Review)]

This work is significant as it provides insights into the global transcriptomic changes of Borrelia burgdorferi during tick feeding. The manuscript also provides methodological advances for the study of the transcriptome of Borrelia burgdorferi in the tick host.

This manuscript documents the study of the transcriptome of Borrelia burgdorferi at 1, 2, 3 and 4 days post-feeding in nymphs of Ixodes scapularis. The authors use antibody-based pull-downs to separate bacteria from tick and mouse cells to perform an enrichment. The data presented support that the transcriptome of B. burgdorferi changes over time in the tick. This work is important as, until now, only limited information on specific genes had been collected. The methodological advances described in this study are valuable for the field.

---

## [Author Response]

The following is the authors' response to the current reviews.

We appreciate the thoughtful critiques of the reviewers. While we agree that performing additional experiments and analyses probing the sensitivity of the technique would be useful for future studies, we are unable to perform additional experiments as our lab has closed. We share this technique as a starting point for further investigation, but it may need to be modified for success in other contexts. We have provided details of the scenarios (life stage, feeding, day, number of ticks) where we successfully sequenced *B. burgdorferi* from ticks, as well as one where we did not (unfed nymphs) as a starting point. We will clarify in proofing that our qPCR experiments show that we capture the vast majority of *B. burgdorferi flaB* mRNA from our input samples, suggesting that we are likely capturing the majority of the *B. burgdorferi*.

In this work, we were most interested in using RNA-seq to perform differential expression analysis between annotated mRNAs across our timepoints. We have provided the number of genes detected in each sample (92% of annotated transcripts on average) as well as the median number of reads covering each gene (604 on average) in the supplemental file containing sequencing statistics. This coverage is highly reproducible across replicates, with an average Pearson correlation of 0.99 between gene expression levels (as Transcripts Per Million) between any two replicates. These data and the fact that many of the gene expression changes we observed align with previous observations of others give us confidence in our differential expression analysis. For those interested in tRNAs or sRNAs, we think that it would be best to modify the protocol to focus specifically on capturing those sequences in the library preparation. We encourage others interested in other aspects of our data to download it and explore it.

We will correct remaining wording issues in proofing.

—————

The following is the authors' response to the original reviews.

Dear Reviewing Editor,

We thank you and the reviewers for the thoughtful comments on our manuscript, and we are excited to submit a revised version of our manuscript “Longitudinal map of transcriptome changes in the Lyme pathogen *Borrelia burgdorferi* during tick-borne transmission.” In response to the reviews, we have made the following changes to our manuscript:

1. We updated the text for increased clarity around experimental details, including statistical analyses.

2. We added additional details about the mapping of non-*Bb* reads as well as more information about *Bb* read coverage.

3. We compared our differentially expressed genes to 4 previous studies of global transcriptional changes in different tick feeding contexts.

4. We updated the discussion to address these comparisons as well as caveats of our study more directly.

Please see our responses to individual comments below.

**Reviewer #1 (Public Review):**
In this study, Sapiro et al sought to develop technology for a transcriptomic analysis of B. burgdorferi directly from infected ticks. The methodology has exciting implications to better understand pathogen RNA profiles during specific infection timepoints, even beyond the Lyme spirochete. The authors demonstrate successful sequencing of the B. burgdorferi transcriptome from ticks and perform mass spectrometry to identify possible tick proteins that interact with B. burgdorferi. This technology and first dataset will be useful for the field. The study is limited in that no transcripts/proteins are followed-up by additional experiments and no biological interactions/infectious-processes are investigated.Critiques and Questions:

We thank the reviewer for these thoughtful critiques and helping us improve our manuscript.

This study largely develops a method and is a resource article. This should be more directly stated in the abstract/introduction.

We edited the abstract and introduction to more directly state that we are sharing a new method and a resource for future investigations. (Lines 29-32; 101-103)

Details of the infection experiment are currently unclear and more information in the results section is warranted. State the species of tick and life-stage (larval vs nymphal ticks) used for experiments. For RNA-seq, are mice are infected and ticks are naïve or are ticks infected and transmitting Borrelia to uninfected mice?

We updated the results section to more clearly state the tick species and life stage and to make it more clear that infected ticks are transmitting *Bb* to naïve mice. (Lines 113-115)

What is the limit of detection for this protocol? Experimental data should be provided about the number of B. burgdorferi required to perform this approach.

We performed this protocol on pools of 6 (for later feeding stages) to 14 (for early stages) infected nymphs. Published studies (PMID: 7485694, PMID: 11682544) suggest that one day after attachment, there may be a few thousand *Bb* per tick, suggesting what we’ve measured here may come from on the order of 104 *Bb*. We were not able to capture consistent data from *Bb* from unfed ticks, which may be due to lower numbers or to an altered transcriptional state caused by lack of nutrients in the unfed tick. We updated the discussion to reflect some of these limitations and uncertainties. (Lines 461-465)

More information regarding RNA-seq coverage is required. Line 147-148 "read coverage was sufficient"; what defines sufficient? Browser images of RNA-seq data across different genes would be useful to visualize the read coverage per gene. What is the distribution of reads among tRNAs, mRNAs, UTRs, and sRNAs?

As we were interested in differential expression analysis, we defined sufficient as the number of reads needed per gene to determine statistically significant expression changes across days, which with DESeq2 is typically 10 reads. We reworded this section for clarity and added additional information about the median number of reads per gene which is also useful in thinking about differential expression analysis. (Lines 163-170) As we chose to focus on differential expression analysis here, we believe these are most relevant metrics to cover.

My lab group was excited about the data generated from this paper. Therefore, we downloaded the raw RNA-seq data from GEO and ran it through our RNA-seq computational pipeline. Our QC analysis revealed that day 4 samples have a different GC% pattern and that a high percentage of *E. coli* sequences were detected. This should be further investigated and addressed in the paper: Are other bacteria being enriched by this method? Why would this be unique to day 4 samples? Does this affect data interpretation?

We appreciate the interest in our data and pointing out this anomaly. We found that the day 4 samples do have a high percentage of reads that mapped to a bacterial species, *Pseudomonas fulva*, rather than ticks as we expected. (The reads that map to *E. coli* also map to *P. fulva*.) We have updated the results to include this information (Lines 156-165). We believe this is likely due to contamination from collecting ticks after they have fallen off mice in cages on day 4, rather than pulling ticks off the mice as in days 1-3. Unfortunately, as our lab has shut down, we cannot investigate the source further. We do think the high percentage of *P. fulva* reads suggests that other bacteria can be enriched with the anti-*Bb* antibody we used. We’ve updated the discussion to highlight this caveat. (Lines 459-460)

While the presence of these bacterial reads did lower our overall *Bb* mapping rate and necessitate deeper sequencing for the day 4 samples, the *Bb* sequencing coverage of these samples is on par with samples from the other days in terms of percentage of genes with at least 10 reads and median number of reads per gene. Fewer than 0.0002% of the reads that map to *Bb* genes in any day 4 sample also map to *P. fulva*. We found that this small fraction of reads is dispersed across 334 genes in which an average of 0.05% (maximally 2.3%) of day 4 reads also map to *P. fulva*. Therefore, these bacterial reads do not change our interpretation of the results comparing gene expression across days, including day 4.

Comprehensive data comparisons of this study and others are warranted. While the authors note examples of known differentially expressed genes (like lines 235-241), how does this global study compare to other global approaches? Are new expression patterns emerging with this RNA-seq approach compared to other methods? What differences emerged from day 1 to day 4 ticks compared to differences observed in unfed to fed ticks or fed ticks to DMC experiments? Directly compare to the following studies (PMID: 11830671; PMID: 25425211; PMID: 36649080).

We added comparisons of our list of DE genes to those noted to change between “unfed tick” and “fed tick” culture conditions (PMID: 11830671 and 12654782), as well as fed nymph to DMC (PMID: 25425211 and 36649080) (Lines 231-252, Figure S4). These comparisons pointed us to two main findings: that global changes to *Bb* in different culture conditions generally agreed with the most dramatic changes we saw in our data, and that the timing of expression increases during feeding may relate to whether genes are more highly expressed in fed ticks or in mammalian conditions. Overall, the majority of our DE genes have been identified in at least one of these studies or in the other studies we compared to outlining RpoS, Rrp1, and RelBbu regulons. As many of these studies were asking slightly different questions and using different conditions and vastly different technology, we would expect some differences to arise from different contexts and some to be purely technical. The genes that were not seen in these previous studies tended to follow the same functional patterns we saw overall, heavily skewing towards genes of unknown function, outer surface proteins, and a handful of genes related to other functions. With the current state of the functional annotation of the genome, it is difficult to assess whether these amount to new expression patterns in and of themselves, so we focused on the overall trends in our data rather than those that were different from other studies.

Details about the categorization of gene functions should be further described. The authors use functional analysis from Drechtrah et al., 2015, but that study also lacks details of how that annotation file was generated. Here, the authors have seemed to supplement the Drechtrah et al., 2015 list with bacteriophage and lipoprotein predictions - which are the same categories they focus their findings. Have they introduced a bias to these functional groups? While it can be noted that many lipoproteins are upregulated (or comment on specific genes classes), there are even more "unknown" proteins upregulated. I argue that not much can be inferred from functional analysis given the current annotation of the B. burgdorferi genome.

We strongly agree that the current annotation of the Bb genome makes it difficult to perform meaningful global functional analysis, but we feel it is useful to get a general overview of gene functions. We described our methods for classifying genes into functional categories in the methods, in which we relied on previously published papers to make our best estimate of gene category (noted for each gene in the Table S4). Due to the lack of annotations for many genes, we focused on the relatively well-defined category of lipoproteins, as these are overrepresented as a group in our upregulated genes, as well as phage genes, which are not necessarily overrepresented, but are still interesting to us. We hope that others will look at the data (particular in Table S4, but also Table S3, or download the raw data and do their own analysis) with their own interests and biases and dig more into genes that we did not highlight specifically. We provide this data as a resource with the hope that some of the genes of unknown function that we see change here will be the subject of future functional studies so that this is less of problem in the future.

**Reviewer #1 (Recommendations For The Authors):**
In general, the paper is well written and digestible for a broad audience. However, some of the figure graphics are unnecessary and take away from the data. Please label tick species and tick life-stage in Figure 1 drawings. The legend of Figure 1 requires citations. The Figure 4B graphic is unnecessary and the colors are confusing as they are too similar to the color palette of Figure 4A, where the colors have meaning. The Figure 5A graphic is unnecessary and takes away from the data embedded within it.

We more clearly labeled the species in Figure 1 and added citations to the legend. We have simplified Figures 4A and 5A for clarity.

Clarify lines 220-259 and Figure 3. What days are being compared? Downregulated genes should also be commented on.

We considered our set of differentially expressed genes as those that changed twofold (multiple hypothesis adjusted p-value < 0.05) in any of the three comparisons shown in Figure 2 (day1 to day2, day1 to day3, day1 to day4). We clarified this at multiple points in the results (i.e Line 273). We commented on downregulated genes throughout, although as there were fewer genes and the magnitude of change was smaller, we focused more on upregulated genes.

Line 327-329, state numbers not percentages. How many Bb proteins were actually detected?

We updated this section to include numbers (Lines 371-374). In concordance with our sequencing data, we found (and were looking for) mainly tick proteins in this experiment.

Data availability: B. burgdorferi and tick oligo sequences used for DASH should be provided in a supplemental table.

We added a supplemental table of these sequences (Table S9). Please note they have been previously published in Dynerman et al. 2020 and Ring et al. 2022.

**Reviewer #2 (Recommendations For The Authors):**
The manuscript is overall well written and easy to follow. The data are compelling and support the conclusions. The discussion of this work is however highly insufficient and needs to be thoroughly edited:- Statistical analysis: The authors mention that DESeq2 was used. Please provide information on the type and the stringency of the tests used for differential gene expression analysis, including any additional potential correction for p-values (Bonferroni). The authors mention that genes with fold changes >2 were used for analysis, yet there is no information on the p-value cut off or if the genes with fold changes >2 were statistically significant. Please provide detail and rationale for the analysis.

We clarified in the results and methods (Lines 200, 642-644) that we required a adjusted p-value < 0.05 from DESeq2’s Wald test with Benjamini-Hochberg correction along with a twofold change when determining our genes of interest. As small fold changes showed statistically significant differences, we chose to set a fold change cutoff in most of our analysis to help us focus on the most highly expressed genes, like other studies we compared our data to. We included all of the DESeq2 results in Table S3 so that others may explore the data with different cutoffs if desired.

- The field has been generating data on gene expression in ticks for decades. Yet, many of these studies are not referenced here. There is no discussion of how the data described here compares to what is known in the literature. For example, Venn diagrams or tables could be included for comparison with the data described lines 208-216. Extensive description and comparison of the data to the literature should be added in the discussion, and similarities/discrepancies should be discussed appropriately.

We added additional comparisons to four different papers looking at global gene expression in *Bb* in the fed tick or tick-like culture conditions (Lines 231-252, Figure S4). This information as well as comparisons to transcriptional regulons (Figure S3) is available in Table S4. In addition to discussing some examples in the results, we added more information in the discussion regarding these comparisons (Lines 420-425). The majority of the genes that we see change over feeding have been previously noted to change expression during the enzootic cycle or be regulated by transcriptional programs active during this timeframe, and we have more clearly stated that. We focused on similarities here as these papers all ask slightly different questions in different contexts and use different technology which could all account for the many differences in individual genes between all of them and our work.

- There is no discussion of the caveats of the study: for example, the authors are using an anti-OspA antibody, which could induce bias. The authors provide in-vitro pull down data supporting that this should not be an issue, but the pull down is performed from BSK-grown bacteria. This caveat should be discussed.

We’ve added a paragraph to the discussion including this caveat and others (Lines 453-463).

- Timing of RNA extraction: There is over 1h of delay between initial tick collection and RNA fixation. The effects of time on gene expression should be discussed.

Although we were able to show that this timeframe did not affect cultured *Bb* gene expression, we added this to the discussion.

- Gene expression is compared to Day 1. This introduces analyses bias as it does not allow identification of transcripts that first change upon initial feeding. This caveat should also be discussed

We added this caveat – that we may miss gene expression changes in the first 24 hours of feeding – to the discussion.

- This study is performed with 1 strain of B. burgdorferi on one tick species. Please provide perspective on the impact of these findings on Lyme disease causing spirochetes and their vectors broadly.

We believe this method could be easily adaptable to study gene expression in other spirochete/vector pairs to determine similarities and differences and we added a comment to the discussion.

- The discussion should also include insights on how to build on this work and include additional areas of method development to increase the recovery of B. burgforferi from ticks or other organisms and facilitate future transcriptomic studies.

We added a few ideas to the discussion noting that this protocol could be modified for use in other timeframes, with other antibodies, or in other organisms. We also highlight the recent advent of TBDCapSeq by Grassmann et al. that may be used in conjunction with this type of protocol.

Minor comments:- Consider re-wording the description of the methods and findings to the third person for coherence.

The majority of the methods are now written in third person.

- Over 90% of the reads did not map to B. burgdorferi: please provide additional information on what these reads mapped to (tick or mouse), and if the data reflects what is known in the literature

We have updated the results and discussion with information about the reads that do not map to *Bb* (Lines 156-166). The majority of reads mapped the tick genome, which is what we expected. While a large number of reads in our day 4 samples unexpectedly mapped to *Pseudomonas fulva*, we do not believe this affects the interpretation of our data as we were still able to get broad genome coverage of *Bb* in these samples.

- Please be more clear in the result section on the life stage of the ticks used for these studies.

We have updated the results to clarify throughout.

- Indicate how many total reads were generated for each sample

This information is present in Table S1.

- Provide statistical analyses for Figures 1C and D.

We added t tests to determine statistical differences for these panels.

**Reviewing Editor (Recommendations for The Authors):**
1. It is important to mention in the abstract (line 27) that 'upregulated genes' is in comparison to day 1. This is also true in the introduction (lines 92-93).

We updated in the results and introduction to more clearly include that day 1 is our baseline measurement.

2. It is also important to discuss in the manuscript that because your 'controls' are day 1 samples, initial transcriptome changes in response to the tick environment might be missed.

This has been added in the discussion as a caveat (Lines 460-463).

3. As someone who does not work with Bb, I would like to have seen a clearer description of what the feeding event looks like. Although there is some text in the introduction that touches on that ('prolonged nature of I. scapularis feeding'), I would like to see something even clearer. Maybe stating that feeding may take from x-y days would clarify that for the non-specialist.

We updated the results to more clearly state that the tick falls off of the mice by around 4 days after feeding, our last time point (Lines 113-115). Additional details of tick feeding are also in the Figure 1 legend.

4. In Fig. 3 linear DNA molecules seem to be drawn to scale. Is that also the case for plasmids? This could be clarified in the legend.

The genome is drawn approximately to scale. We noted this and updated the legend with more information about how linear and circular plasmid names denote their size.

5. Figure 5C: Colors are a bit confusing here. The legend indicates that they refer to fold changes, but the scale in the panel shows expression levels, not fold changes. Please clarify. Also, is this really TPM or RPKM? If comparisons of relative levels between different genes are made, number of reads should be normalized by gene length.

The heatmap in Figure 4C does show expression levels, and we updated the legend to more clearly state this. The highlighted gene names are meant to show which genes change twofold during this time (those present in panel A). The data are presented as TPM (transcripts per million), which, like RPKM, is normalized by gene length (PMID: 20022975).